# READY2UNLEARN: A LEARNING-TIME APPROACH FOR PREPARING MODELS WITH FUTURE UNLEARNING READINESS

## ABSTRACT

This paper introduces **Ready2Unlearn**, a learning-time optimization approach designed to facilitate future unlearning processes. Unlike the majority of existing unlearning efforts that focus on designing unlearning algorithms, which are typically implemented reactively when an unlearning request is made during the model deployment phase, Ready2Unlearn shifts the focus to the training phase, adopting a *"forward-looking"* perspective. Building upon well-established meta-learning principles, Ready2Unlearn proactively trains machine learning models with *unlearning readiness*, such that they are well prepared and can handle future unlearning requests in a more efficient and principled manner. Ready2Unlearn is model-agnostic and compatible with any gradient ascent-based machine unlearning algorithms. We evaluate the method on both vision and language tasks under various unlearning settings, including class-wise unlearning and random data unlearning. Experimental results show that by incorporating such preparedness at training time, Ready2Unlearn produces an *unlearning-ready* model state, which offers several key advantages when future unlearning is requested, including reduced unlearning time, improved retention of overall model capability, and enhanced resistance to the inadvertent recovery of forgotten data. We hope this study could inspire future work to explore more proactive strategies for equipping machine learning models with built-in readiness towards more reliable and principled machine unlearning.

## 1 INTRODUCTION

Machine unlearning (Cao & Yang, 2015) refers to the process of removing the imprint left by specific data samples during the training of a machine learning model. AI developers employ machine unlearning for various purposes. In the context of privacy protection, it is often necessary to remove the influence that individuals' personal data has had on a model's learned parameters (Jang et al., 2023; Miranda et al., 2024; Zhou et al., 2023). Legal frameworks such as the European Union's General Data Protection Regulation (GDPR) (Protection, 2018) and the California Consumer Privacy Act (CCPA)[1] grant individuals the right to control their personal data, including revoking it from organizations that use the data to train models. Beyond privacy, machine unlearning is also used to address ethical and security concerns by removing the influence of harmful or sensitive data (Schoepf et al., 2024; Yu et al., 2023), such as preventing large language models (LLMs) from retaining information that could be misused for developing bioweapons or launching cyberattacks (Barrett et al., 2023; Sandbrink, 2023; Li et al., 2024). Additionally, unlearning can be applied to improve model performance by eliminating the impact of low-quality or noisy training samples (Cao & Yang, 2015; Wang et al., 2023). These diverse applications emphasize that machine unlearning is important and practically meaningful.

Numerous unlearning algorithms have been introduced in recent years, employing techniques such as preference optimization (Zhang et al., 2024b; Mekala et al., 2025), gradient rectification (Hoang et al., 2024; Lin et al., 2024; Wang et al., 2025b), and data augmentation (Peng et al., 2025; Cha et al., 2024; Mekala et al., 2025), to name a few. Despite these efforts, unlearning remains a chal-

---

[1]https://oag.ca.gov/privacy/ccpa

lenging task. First, it often requires considerable time or a large number of optimization steps to achieve satisfactory forgetting, especially for large-scale models such as LLMs (Eldan & Russinovich, 2023). Additionally, balancing the trade-off between forgetting data and preserving overall model utility is difficult (Hoang et al., 2024; Lin et al., 2024; Wang et al., 2025b), as the unlearning process can lead to catastrophic forgetting (Zhang et al., 2024a). Moreover, some studies suggest that current unlearning methods may not be as reliable as they appear, with data that seems to be "forgotten" often being easily recovered (Kim et al., 2025; Zhang et al., 2024c; Hong et al., 2024).

Such long-standing challenges prompt us to ask: *Is the model truly ready to forget when unlearning is initiated, and can we take steps during training to proactively prepare it with unlearning readiness against potential future unlearning requests?* In this paper, we explore the possibility that equipping the model with unlearning readiness during the training phase to benefit the unlearning process that may take place later after model deployment, with improved efficiency and reliability.

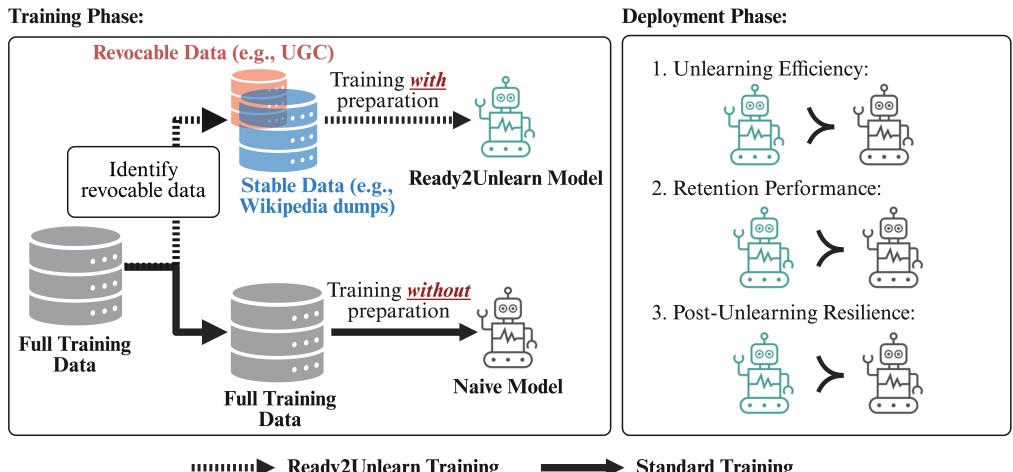

Figure 1: Comparison of learning with (top) and without (bottom) unlearning preparation.

The problem is depicted in Figure 1. We make a practical assumption that, in real-world applications, not all training data is equally likely to be subject to future unlearning requests (Krishnan et al., 2025). Some data, such as user-generated content (UGC) (De Min et al.), are more likely to be revoked due to privacy or other regulatory concerns, while other data, like public datasets (e.g., Wikipedia dumps), are less likely to trigger such requests.

Motivated by this observation, we introduce a method (Ready2Unlearn) that equips the model, during training, with an awareness of potential unlearning actions that may occur later during deployment for data that are more likely to be revoked, which we term *revocable data* throughout the manuscript. The remaining data are therefore termed *stable data*, reflecting their inherent stability, that is, their low likelihood of being subject to future revocation requests. In terms of which data should be classified as revocable or stable, we consider this to be largely a managerial decision, guided by organizational priorities and regulatory considerations. For example, in many enterprise settings subject to GDPR, customer data are routinely classified by their purpose (e.g., marketing, onboarding) and source (e.g., publicly accessible information, user-provided data, system-derived records, or behavioral inferences).[2] Such operational data-governance practices naturally provide insight into which categories are more likely to be revoked (e.g., user-provided data) and which are generally stable (e.g., publicly accessible information). In this work, we do not prescribe a one-size-fits-all algorithmic approach for this categorization, as it is best determined by practitioners based on domain-specific knowledge and institutional policies.

Our expectation is that, once a model has undergone learning with Ready2Unlearn and is deployed in the environment, and an unlearning request arises for the revocable data, it can 1) unlearn more

---

[2] See *GDPR deep dive—how to implement the 'right to be forgotten'*, available at: `https://www.bankinghub.eu/finance-risk/gdpr-deep-dive-implement-right-forgotten?utm_source=chatgpt.com`, Fig. 3.

efficiently with fewer unlearning steps, 2) better preserve performance on the unaffected stable data, and 3) exhibit greater post-unlearning robustness against the inadvertent recovery of *previously forgotten data*, compared to a model trained without unlearning readiness (the naïve model). Two points to note here: First, by "learning", we refer to either training from scratch or further fine-tuning (as commonly done in the LLM context). Second, by "unlearning", we mean any gradient ascent-based unlearning algorithms. The reason that only gradient ascent-based unlearning can benefit from our approach is that, as later described in Section 3, Ready2Unlearn modifies the model parameters during training toward a state that is particularly amenable to subsequent gradient ascent-based parameter updates. As such, the model is "*pre-conditioned*" to respond more efficiently and reliably, exclusively when gradient ascent is later applied to remove specific data.

Ready2Unlearn adopts meta-learning principles, particularly the MAML algorithm (Finn et al., 2017). Traditional meta-learning techniques aim to find a model initialization such that performing a small number of gradient *descent* steps from this starting point allows the model to quickly adapt to new tasks. In our setting, we aim to obtain a model "initialization" (the prepared model state) such that, based on this initialization, applying only a few gradient *ascent* steps (representing unlearning operations) leads to a significant increase in loss for the intended forgotten data, while preserving overall model utility and ensuring strong post-unlearning resilience.

To demonstrate preparing models for unlearning readiness at the learning stage is feasible, we test the proposed Ready2Unlearn idea on both vision and language unlearning tasks, including class-wise unlearning and random data unlearning. In our experiments, we show that with Ready2Unlearn preparation, the model responds much more quickly to unlearning requests compared to models without such preparedness or baseline approaches. Additionally, the model's overall utility is better preserved, with reduced risk of catastrophic forgetting. Furthermore, when attempting to recover the forgotten data by further fine-tuning the unlearned model on data with similar distributional characteristics, such as stylistically or semantically related examples, the model exhibits greater resistance to recovery compared to its unprepared counterpart. We hope our findings offer fresh perspectives in machine unlearning and additionally inspire future work on incorporating such "looking-ahead" designs in broader algorithmic contexts.

## 2 RELATED WORK

**Machine unlearning.** Due to the growing need to eliminate traces of specific data from machine learning models throughout the AI lifecycle, machine unlearning techniques have seen rapid advancement (Liu et al., 2024b; Wang et al., 2024a). Current research in this area primarily focuses on developing effective unlearning algorithms (Peng et al., 2025; Lin et al., 2024; Jia et al., 2024; Ji et al., 2024; Zhou et al., 2023; Jang et al., 2023; Fan et al., 2023; Chen & Yang, 2023), designing rigorous evaluation protocols (Liu et al., 2025; Kim et al., 2025; Zhang et al., 2024c; Shi et al., 2024; Hong et al., 2024), and identifying and addressing practical, real-world challenges (Shen et al., 2024), such as the unavailability of users' erased data during the unlearning process (Wang et al., 2025a), which relaxes some classic assumptions in traditional unlearning practices. While existing machine unlearning methods generally perform well in typical scenarios, they often face critical limitations, such as prolonged unlearning time (Eldan & Russinovich, 2023), catastrophic forgetting (Peng et al., 2025; Wang et al., 2025b; Lin et al., 2024; Hoang et al., 2024; Choi et al., 2024), and vulnerability under more rigorous evaluation conditions, such as susceptibility to jailbreak attacks (Zhao et al., 2024a) and the ease with which forgotten data can be recovered (Kim et al., 2025; Hong et al., 2024; Zhang et al., 2024c). This suggests that current unlearning solutions are inadequate and still have considerable room for improvement. Our work contributes to this line of research by demonstrating that it is possible to prepare models at the learning stage with future unlearning readiness to further enhance the efficiency, reliability, and robustness of existing unlearning methods. We believe this offers a new perspective for enhancing current unlearning practices.

**Training-time regularization to mitigate memorization.** Mitigating training data memorization is a reasonable approach to alleviate future unlearning efforts, as models that memorize less are less tied to specific details of data points, making it easier to forget them later (Zhao et al., 2024b). This is typically achieved through regularization techniques applied during model training. Well-known examples include dropout (Srivastava et al., 2014), weight decay (Ng, 2004; Krogh & Hertz, 1991), data augmentation (Shorten & Khoshgftaar, 2019), and differentially private learning (Abadi et al.,

2016), to name a few. In addition to these general, model-agnostic methods, there are more specialized techniques, such as Goldfish regularization (Hans et al., 2024), which randomly excludes tokens from loss computation, and NEFTune (Jain et al., 2023), which adds noise to embedding vectors, both tailored for large language models. At a broader level, the Ready2Unlearn method presented in this work can also be positioned within the regularization literature. Furthermore, it adds to this line of research with a novel unlearning-specific regularization technique. Ready2Unlearn is particularly advantageous in machine unlearning scenarios, where existing general regularization techniques often fall short, as they are, by nature, not optimized for unlearning contexts. Thus, we believe this work provides a more targeted solution from this perspective.

**Meta-learning.** Another relevant line of research is meta-learning, also known as "learning to learn", which aims to train models that can rapidly adapt to new tasks with limited data or computational resources (Hospedales et al., 2021). A notable example is model-agnostic meta-learning (MAML) (Finn et al., 2017), which seeks to find a model initialization such that only a few gradient updates are required for effective adaptation to new tasks. This paradigm has been widely adopted across various fields, such as domain generalization (Li et al., 2018), safeguarding LLMs against adversarial attacks (Tamirisa et al., 2024), and hyperparameter optimization (Baik et al., 2020). Our work draws inspiration from the core idea of MAML, but shifts the objective from fast adaptation to new tasks to fast, reliable machine unlearning. By incorporating a meta-objective at training time, Ready2Unlearn optimizes the model into an *unlearning-ready* state—a parameter configuration from which future gradient ascent updates (representing unlearning) can proceed in a well-behaved and principled manner. This preparation leads to several desirable properties, including improved unlearning efficiency, better retention of overall model capability, and increased resistance to the reintroduction of forgotten data. We are aware of prior work that also applies meta-learning techniques in the context of machine unlearning (Huang et al., 2024). However, our approach differs fundamentally in both the timing and the goal of applying meta-learning: their method adopts meta-learning during the unlearning phase, after model deployment, whereas Ready2Unlearn introduces meta-learning at training time, proactively preparing the model before any unlearning request arises. This distinction positions our approach as a *preemptive* strategy, marking a conceptual departure from the majority of reactive unlearning methods toward a proactive paradigm. In this sense, we believe that Ready2Unlearn introduces the meta-learning idea to the field of machine unlearning with a novel use case (i.e., shifting reactive unlearning to a proactive mindset).

## 3 LEARNING WITH UNLEARNING PREPAREDNESS

### 3.1 UNLEARNING PROCESS

We assume that the model developer has access to a dataset $\mathcal{D} = \mathcal{D}_{\mathrm{f}} \cup \mathcal{D}_{\mathrm{r}}$, which comprises both revocable data (likely to be unlearned in the future, referred to as *forget data*[3], $\mathcal{D}_{\mathrm{f}}$) and stable data (unlikely to be unlearned, referred to as *retain data*, $\mathcal{D}_{\mathrm{r}}$), and builds a model with weights $\theta_P$, where a preparation P has been applied. Our goal is to design P such that $\theta_P$ performs well on three metrics when future unlearning is triggered: `efficiency_metric`$(\theta_P)$, `retention_metric`$(\theta_P)$, and `resistance_metric`$(\theta_P)$. In this work, by "unlearning", we refer to the process of applying gradient ascent steps to adjust $\theta_P$ based on the forget data. Gradient ascent is widely used as an effective unlearning strategy due to its simplicity and its model- and data-agnostic nature (Yao et al., 2023; Maini et al., 2024; Graves et al., 2021; Hoang et al., 2024; Li et al., 2023a; Tarun et al., 2023; Thudi et al., 2022; Ji et al., 2024). Moreover, we assume that the retain data is not accessible during unlearning. We impose this stricter condition because, in many real-world scenarios, access to retain data is often impossible, and retraining the model on this data can be prohibitively costly (Wang et al., 2024b; Cheng et al., 2024; Foster et al., 2024). Thus, throughout the paper, unlearning specifically refers to a clean process where gradient ascent steps are applied solely to the forget data.

### 3.2 PROBLEM FORMULATION AND METRICS

Let `GA` denote the gradient ascent unlearning operation, which maps the prepared model $\theta_P$ to the *unlearned model* $\theta'_P = $ `GA`$(\theta_P; \mathcal{D}_{\mathrm{f}})$. Let `RC` denote the recovery operation, which further fine-tunes

---

[3]Here, in presenting the methodology, we use the terms "forget data" and "retain data" to denote revocable and stable data, respectively, to align with the conventional terminology in the machine unlearning literature.

the unlearned model $\theta'_P$ on data samples $\mathcal{D}_{rc}$, similar in style to the forget data, producing the *post-recovery model* $\theta''_P = \mathtt{RC}(\theta'_P; \mathcal{D}_{rc})$. Below, we define three key metrics.

**Efficiency metric.** We say a preparation P leads to efficient unlearning if the model $\theta_P$ experiences a substantial increase in loss on the forget data after only a few, or even a single, gradient ascent unlearning update. Thus, $\mathtt{efficiency\_metric}$ is defined as the loss (e.g., classification error) of the unlearned model on the forget data. A higher loss indicates greater unlearning efficiency.

**Retention metric.** We say a preparation P enables strong capability retention if the unlearned model $\theta'_P$ preserves much of its performance on the retain data. Thus, $\mathtt{retention\_metric}$ is defined as the unlearned model's performance (e.g., classification accuracy) on the retain data. Higher performance signifies stronger retention.

**Resistance metric.** We say a preparation P equips the model with greater resistance to the inadvertent recovery of erased data if, upon further fine-tuning the unlearned model on data similar in style to the forget data,[4] the resulting model is less likely to regain information about the forgotten data. Thus, $\mathtt{resistance\_metric}$ is defined as the loss of the post-recovery model $\theta''_P$ on the forget data. A higher loss indicates stronger post-unlearning resilience.

### 3.3 Unlearning-Ready Training

To proactively prepare models during learning towards more efficient and principled unlearning in the future, we introduce Ready2Unlearn, a *forward-looking* method outlined in Algorithm 1 in Appendix A. Inspired by meta-learning, this approach prepares the model during learning to be *unlearning-ready*, ensuring that future unlearning via gradient ascent behaves in a stable and reliable manner. At a high level, we learn the model $\theta_P$ with unlearning preparedness using a *dual-loop* optimization structure inspired by MAML (Finn et al., 2017), comprising an *inner-loop* gradient update and an *outer-loop* optimization.

**Method intuition.** The key idea behind Ready2Unlearn is to *optimize for the future*. Rather than maximizing the model's immediate performance, we simulate potential unlearning operations that may occur later and optimize the model to be ready for them. Specifically, this is achieved by designing the inner-loop gradient update to *mimic* unlearning actions, representing the "unlearner's" first move. In the outer loop, the model parameters are optimized to maximize three desirable properties—efficiency, retention, and resistance—against the unlearner's move simulated in the inner loop. This forward-looking optimization ensures that, when unlearning is eventually triggered, the model will exhibit optimal performance in terms of unlearning efficiency, capability retention, and post-unlearning resilience. Essentially, we are preparing the model for future unlearning challenges rather than simply optimizing for the current task. This forward-looking perspective distinguishes our method from conventional approaches. We illustrate this forward-looking nature using conceptual 1D loss landscapes in Figure 2.

**Unlearning-ready objective.** Let $\mathcal{L}$ denote the loss function suitable for the task at hand. We now describe the optimization objective of Ready2Unlearn.

To *enhance unlearning efficiency* (i.e., to achieve fast unlearning), we seek to find a $\theta$ upon which even a single step of gradient ascent (representing an unlearning step), $\mathtt{GA}(\theta)$, leads to a substantial increase in loss on the forget data. Thus, we consider maximizing $\mathcal{L}(\mathtt{GA}(\theta); \mathcal{D}_f)$.

To *avoid catastrophic forgetting* (i.e., to promote performance retention), we aim to identify a $\theta$ such that, after unlearning, the resulting parameters, $\mathtt{GA}(\theta)$, still enable the model to preserve much of its performance on the retain data. Accordingly, we consider minimizing $\mathcal{L}(\mathtt{GA}(\theta); \mathcal{D}_r)$.

To *achieve more reliable unlearning* (i.e., to improve post-unlearning resilience), we aim to find a $\theta$ such that, after unlearning with $\mathtt{GA}(\theta)$ and further fine-tuning the unlearned model on recovery data (which shares a similar style to the forget data), the resulting model does not regain significant information about the forgotten data. We operationalize this by minimizing $\mathcal{L}(\mathtt{GA}(\theta); \mathcal{D}_{rc})$. The rationale behind is to direct the unlearning process so that it removes the most distinctive characteristics of the forget data, rather than merely erasing superficial patterns that may also be present in the recovery data. Otherwise, if the model only unlearns superficial patterns, the loss on the re-

---

[4]Please refer to Appendix J, where we demonstrate how the similarity can be measured.

covery data is likely to increase as well, which is exactly what we aim to prevent by minimizing the aforementioned loss.

Put together, our objective is to solve the following optimization problem to obtain the unlearning-ready model $\theta_P$:

$$\min_{\theta}[-\mathcal{L}(\texttt{GA}(\theta), \mathcal{D}_{\mathrm{f}}) + \lambda_1 \cdot \mathcal{L}(\texttt{GA}(\theta), \mathcal{D}_{\mathrm{r}}) + \lambda_2 \cdot \mathcal{L}(\texttt{GA}(\theta), \mathcal{D}_{\mathrm{rc}}) + \lambda_3 \cdot \mathcal{L}(\theta; \mathcal{D}))], \qquad (1)$$

where $\lambda_1$, $\lambda_2$, and $\lambda_3$ are scalar weights for the respective losses. The first three terms represent the *"future objectives"*, ensuring that when future unlearning takes place, their respective objectives will be optimized accordingly, reflecting the forward-looking nature of Ready2Unlearn. The final term serves as the *"current objective"*, optimizing the model's current utility prior to any unlearning action being taken. Please refer to Algorithm 1 in Appendix A for the full optimization procedure. It should be noted that the additional meta-objectives may introduce extra computational overhead compared to standard training. We discuss this in Appendix D.

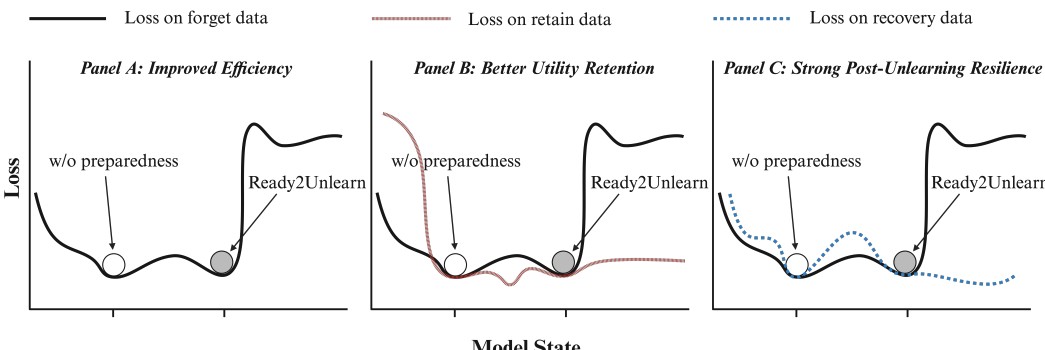

Figure 2: An illustration of the forward-looking nature of Ready2Unlearn using conceptual 1D loss landscapes. **Panel A:** The model state obtained with unlearning preparedness (gray circle) lies adjacent to a steep ascent in the loss landscape with respect to the forget data, such that even a single gradient ascent step can trigger a substantial loss increase, enabling fast and effective unlearning. In contrast, the unprepared model (white circle) lies in a flatter region far from any steep increase, requiring many steps to achieve comparable unlearning. **Panel B:** The model state optimized by Ready2Unlearn resides in a region where the retain-data loss remains low and stable despite gradient ascent updates on forget data, thereby preserving the model's utility. The unprepared model resides in a region where unlearning actions (i.e., gradient ascent on forget data) adversely impact performance on retain data, as indicated by a sharp increase in retain loss. **Panel C:** Around the unprepared model state, the loss landscapes of forget data and recovery data exhibit high similarity. As a result, fine-tuning on recovery data inadvertently lowers the forget loss as well, undoing the unlearning. In contrast, Ready2Unlearn prepares the model in a region where the recovery loss is low and exhibits a distinct pattern from the forget loss, thereby making the model less likely to reacquire forgotten information during further fine-tuning on recovery data.

## 4 EXPERIMENTS

### 4.1 CLASS-WISE UNLEARNING IN IMAGE CLASSIFICATION

**Experiment setup.** We consider image classification tasks using the MNIST (Deng, 2012) and PathMNIST (Yang et al., 2023; 2021) datasets. Each class is treated in turn as the revocable data (i.e., forget data, $\mathcal{D}_{\mathrm{f}}$), prepared at training time for potential future unlearning requests. The remaining classes are treated as stable data (i.e., retain data, $\mathcal{D}_{\mathrm{r}}$), used to evaluate the model's capability retention after unlearning.[5] Following prior unlearning research (Zhou et al., 2023), we use a convolutional neural network (CNN) as the classifier. The model is trained using the negative

---

[5]We additionally investigate in Appendix F the case where the forget and retain sets have overlap, and in Appendix H how the method scales with forget-data diversity.

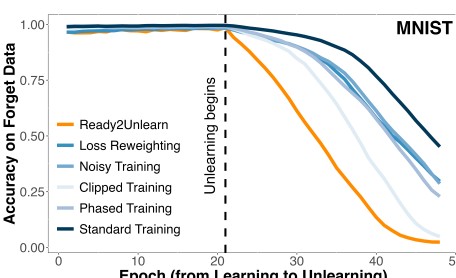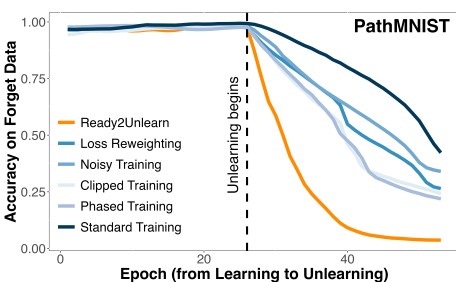

Figure 3: Comparison of unlearning efficiency for MNIST (left) and PathMNIST (right). Each line represents the average forget-data accuracy across all class-wise unlearning settings, where each class is treated as the forget class in turn. All methods are evaluated with the same unlearning rate of $1 \times 10^{-5}$ for a fair comparison. The vertical dashed line marks the moment when unlearning begins. Statistical comparisons reveal that, for both datasets, at the point when the best-performing method (Ready2Unlearn) first falls below random guessing performance on the forget data, its corresponding accuracy is significantly lower than that of the best competing baseline approach (both $p < 1 \times 10^{-17}$, one-tailed t-test, based on 10 independent trials).

log-likelihood loss function, and performance is evaluated using classification accuracy. We apply a first-order approximation to compute the meta-gradients (as detailed in lines 6, 7, and 8 in Algorithm 1). We use an inner-loop unlearning rate $\alpha$ of $1 \times 10^{-5}$, an outer-loop learning rate $\eta$ of $2 \times 10^{-4}$, and set the loss weights $\lambda_1$ and $\lambda_3$ to 2 and 4, respectively,[6] during training. In the deployment phase, when unlearning is executed, we apply gradient ascent on the forget data with a step size of $1 \times 10^{-5}$.

**Baseline methods.** We consider several baseline methods that reduce the training imprint of revocable data, enabling more efficient future unlearning. *Standard Training* serves as the basic approach, where the model is trained using SGD without distinguishing between revocable and stable data. *Loss Reweighting* reduces the influence of revocable data during training by assigning it half the loss weight of stable data (Kendall et al., 2018). *Noisy Training* perturbs revocable images by adding standard Gaussian noise scaled by 0.3, which helps to prevent the model from overly memorizing these examples and supports easier unlearning in the future (Rifai et al., 2011; Shorten & Khosh-goftaar, 2019). In *Clipped Training*, gradient clipping is applied to revocable data to constrain its influence on model parameters, which may help prevent overfitting and support easier unlearning (Mikolov et al., 2012). Finally, *Phased Training* starts with training on the full dataset in the first half of the training period and then proceeds to train only on stable data in the second half, allowing the model to initially learn from revocable data without continued exposure, which may ease future unlearning (Bengio et al., 2009; Goodfellow et al., 2013).

**Results and discussion.** We present the unlearning efficiency benchmarking results in Figure 3, which lead to several key observations. First, when the model is trained without consideration for future unlearning, the unlearning process is notably slow. This is evident from the Standard Training baseline, where the accuracy on the forget data remains consistently higher than that of all other methods, which incorporate varying degrees of unlearning preparedness. This underscores the importance of training-time preparation for enabling more efficient unlearning later. Second, among the evaluated methods, Ready2Unlearn exhibits the highest unlearning efficiency. We observe that once unlearning begins, the model prepared with Ready2Unlearn immediately undergoes a sharp decline in accuracy on the forget data, reflecting a quick response compared to other models. For example, when the model equipped with Ready2Unlearn reaches 50% accuracy, the other models still maintain a relatively high accuracy of around 75% on average. Third, at the moment when unlearning is initiated (marked by the vertical dashed line), the accuracies on the forget data are comparable across all methods, indicating that the superior efficiency of Ready2Unlearn is not at the cost of suboptimal performance on the forget data prior to unlearning. In other words, the unlearning readiness enabled by our approach does not significantly compromise the model's pre-

---

[6]Throughout all experiments in this work, the loss weight parameters were selected based on a held-out validation set, following the principle that the chosen weights should enable the fastest unlearning while ensuring that the training performance does not drop by more than 1% relative to the same model trained without meta-objectives. For a detailed ablation analysis of the loss weights, please refer to Appendix E.

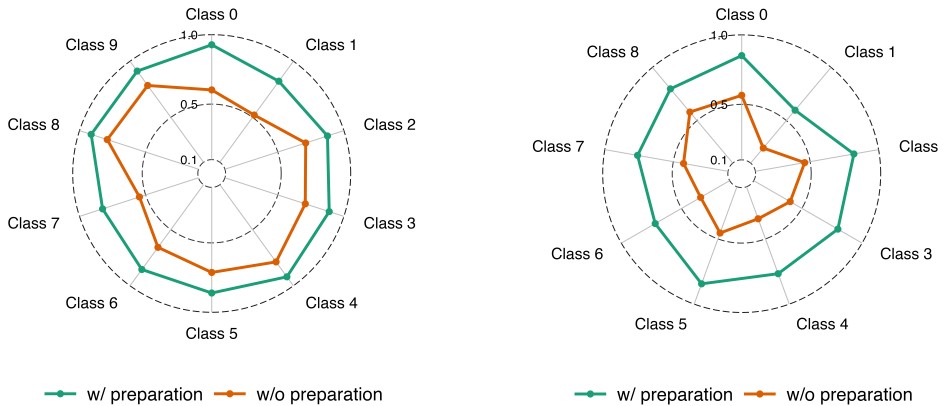

Figure 4: Performance retention for MNIST (left) and PathMNIST (right). Each axis of the radar chart corresponds to a class treated as the forget class. The value on each axis shows the model's retain accuracy when its forget accuracy reaches random guessing. Statistical tests confirm that, for both datasets and across all forget classes, the accuracy on retain data achieved with preparation is significantly higher than that achieved without preparation (all $p < 1 \times 10^{-14}$, one-tailed t-test, based on 10 independent trials).

unlearning performance. Overall, the results demonstrate that training a model with unlearning in mind improves the unlearning process at a later stage, with more tailored approaches, such as Ready2Unlearn, offering greater advantages. We also examine how the duration and timing of preparatory steps with Ready2Unlearn affects future unlearning efficiency; please see Appendix C and Appendix I for details.

We evaluate whether Ready2Unlearn equips the model with better capability retention when unlearning is performed. Specifically, we examine the model's accuracy on the retain data after unlearning has driven the forget class accuracy down to the level of random guessing. The results visualized in Figure 4 reveal that models trained with Ready2Unlearn consistently maintain substantially higher accuracy on retain data compared to those trained without unlearning preparation (i.e., Standard Training). It is important to note that during unlearning, we assume retain data is inaccessible; thus, unlearning is performed solely by applying gradient ascent to the forget data, without any concurrent training on retain data as is often done in prior work (Huang et al., 2024). Thus, the improved retention of performance is entirely attributed to the training-time preparation, emphasizing the advantage of Ready2Unlearn in handling more complex unlearning scenarios where the retain data is inaccessible, with a forward-looking design.

## 4.2 RANDOM DATA UNLEARNING IN TEXT GENERATION

**Experiment setup.** We use LLaMA-3.2-1B[7] as the target model and benchmark unlearning efficiency on two widely adopted unlearning corpora: MUSE-Books and MUSE-News (Shi et al., 2024). We use their "raw" data splits and adhere to the official forget/retain partitions for both datasets. To evaluate whether Ready2Unlearn improves post-unlearning resilience, we use a separate dataset, the Enron email corpus[8], which offers an ideal setting for recovery evaluation due to the strong stylistic consistency across email messages. We randomly split the data into three subsets: one for $\mathcal{D}_f$, one for $\mathcal{D}_r$, and one ($\mathcal{D}_{rc}$) for further fine-tuning the unlearned model to assess potential recovery of forgotten information. Throughout the experiments, by learning, unlearning, and recovering, we refer to fine-tuning the model with the next-word prediction objective using cross-entropy loss (also used as the performance evaluation metric). For unlearning efficiency evaluation, we set the loss scaling factors $\lambda_1$ and $\lambda_3$ to 2 and 4, respectively. For post-unlearning resilience evaluation, we set $\lambda_2 = 3$ and $\lambda_3 = 4$. We apply gradient ascent unlearning with a step size of $1 \times 10^{-6}$.

---

[7]https://huggingface.co/meta-llama/Llama-3.2-1B
[8]https://www.cs.cmu.edu/~enron/

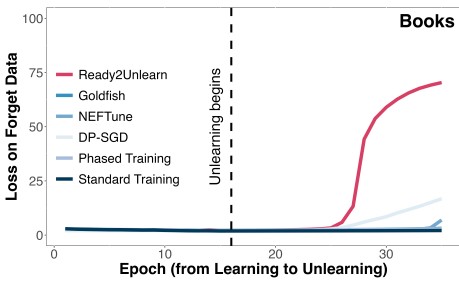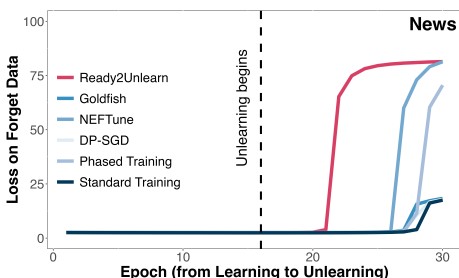

Figure 5: Comparison of unlearning efficiency for MUSE-Books (left) and MUSE-News (right) using Llama-3.2-1B as the target model. Each line represents the cross-entropy loss on the forget data for each method. All methods are evaluated with the same unlearning rate of $1 \times 10^{-6}$ for a fair comparison. The vertical dashed line marks the moment when unlearning begins. Statistical testing reveals that, for both datasets, at the Epoch when the forget loss of Ready2Unlearn first crosses 20, this loss is significantly larger than that achieved by the best competing baseline method (both $p < 1 \times 10^{-19}$, one-tailed t-test, based on 10 independent trials).

**Baseline methods.** We consider the following baseline methods. *Standard Training* and *Phased Training* are included as defined in the earlier class-wise unlearning setup. We additionally consider *DP-SGD* (Li et al., 2021; Abadi et al., 2016), which fine-tunes the model with differential privacy using a clipping norm of 0.1. We include two language model-specific techniques: *Goldfish* (Hans et al., 2024), which randomly excludes tokens from the loss computation with a probability of 0.25, and *NEFTune* (Jain et al., 2023), which injects noise into the embedding vectors with a scaling factor of $\alpha = 5$. These techniques are applied to revocable data during training to mitigate over-memorization, thereby enabling more efficient unlearning later on.

**Results and discussion.** We compare the unlearning efficiency of all methods in Figure 5. The results indicate that with a very small gradient ascent step size during unlearning (as in this experiment, $1 \times 10^{-6}$), achieving sufficient forgetting for LLMs typically necessitates a considerable number of optimization steps—meaning that the effects of unlearning are not immediately apparent upon initiation. Fortunately, consistent with the findings from the class-wise unlearning experiments, incorporating preparatory steps during training can effectively reduce the time needed to achieve meaningful unlearning later on, without significantly compromising the model's performance prior to unlearning. Notably, models trained with Ready2Unlearn begin to forget earlier than all baselines and reach noticeable unlearning with fewer gradient ascent steps, demonstrating that our approach generalizes well to transformer-based language models, beyond classic deep neural networks like CNNs. Additional evaluation results using Llama-3.2-3B[9], GPT-2, and GPT-2 Medium (Radford et al., 2019) as target models are provided in Appendix B.

Figure 6 compares the post-unlearning resilience of models trained with and without Ready2Unlearn preparation. The most notable observation is that, after the loss has plateaued following further fine-tuning the unlearned model on a dataset similar in style to the forget data, the model prepared with Ready2Unlearn consistently maintains a higher loss on the forget data compared to the model without preparation. This indicates that the Ready2Unlearn-prepared model is more resistant to regaining the forgotten information—even when exposed to data with similar characteristics—than its non-prepared counterpart. This resilience arises from the gradient term in line 8 in Algorithm 1 in Appendix A, which directs the model to unlearn the most distinctive, data-specific features of the forget data, rather than merely suppressing superficial patterns that could easily re-emerge during subsequent (inadvertent) fine-tuning. In contrast, without this preparatory step, the model is more susceptible to relearning those superficial patterns, leading to a lower loss on the forget data after fine-tuning. See Figure 7 for an illustration, and Appendix G for additional evidence from the lens of representation analysis. Thus, from this perspective, we believe Ready2Unlearn generates new insights into more targeted machine unlearning (Liu et al., 2024a), where the information to be removed from the model is much more nuanced and selective.

---

[9]https://huggingface.co/meta-llama/Llama-3.2-3B

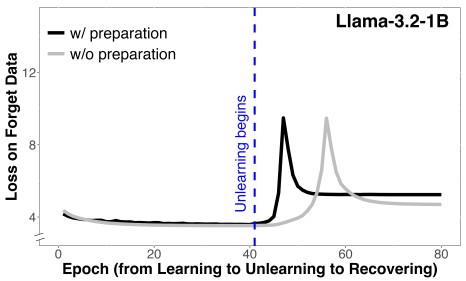 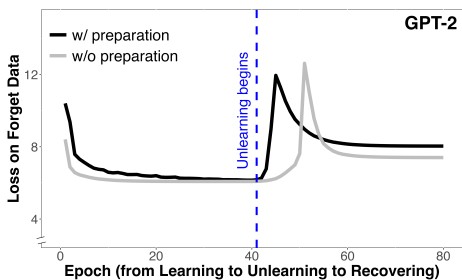

Figure 6: Loss on forget data across three phases for Llama-3.2-1B (left) and GPT-2 (right). The loss decreases and plateaus during training, rises sharply during unlearning, and then decreases and plateaus again during recovery via further fine-tuning on stylistically similar data. Statistical analysis shows that, for both models, at Epoch 70, the forget loss achieved with preparation is significantly larger than that achieved without preparation (both $p < 1 \times 10^{-12}$, one-tailed t-test, based on 10 independent trials).

```
1. login:  pallen pw: ke9davis I do not  think these are required by the ISP 2. static IP address
1. login:  pallen pw: ke9davis I do not  think these are required by the ISP 2. static IP address
```

Figure 7: A visual example from the forget set comparing token-level loss after unlearning a model trained without preparation (Standard Training, top) and with Ready2Unlearn preparation (bottom). Color shading indicates relative loss per token, with darker tones representing higher values. In the standard case, loss is spread more uniformly across tokens, suggesting that the model treats all content equally during unlearning rather than prioritizing the removal of more critical information. In contrast, the model prepared by Ready2Unlearn tends to assign higher loss to more distinctive, data-specific tokens, such as login names or passwords (e.g., "pallen", "ke9davis"), showcasing its focus on unlearning more meaningful, sensitive information rather than generic content. This makes the removed information harder to recover without access to the original forget data.

## 5 CONCLUSION

In this paper, we introduce Ready2Unlearn, a forward-looking approach that proactively prepares neural network models during training to enhance their readiness for future unlearning. By incorporating preemptive steps into the learning process, our method enables models to unlearn more efficiently while preserving much of their utility and exhibiting greater resilience after unlearning. We demonstrate that unlearning should not only be treated as an afterthought but rather as a critical aspect of model lifecycle management that can be proactively addressed through forward-looking training designs. We believe this work offers a new perspective for addressing challenges posed by evolving data governance and privacy demands, particularly in the context of machine unlearning.

## 6 LIMITATIONS

Our work also has limitations that can be improved in future research. First, our study simplifies the revocation risk categorization of training data into two groups: revocable data and stable data. This binary classification may not capture the full spectrum of revocation risks. Future work could explore more granular risk levels or even continuous risk gradations (by estimating the likelihood of future unlearning) in training data, moving beyond the basic dichotomy. Second, Ready2Unlearn does not provide incremental benefits for data whose revocation status was incorrectly forecasted; such misjudged instances would be handled in the same manner as in standard reactive unlearning methods. Developing efficient methods to estimate the likelihood of future unlearning requests is therefore a promising and practical direction for future research. Third, our method relies on gradient-ascent-based unlearning algorithms. While gradient-ascent is predominantly used and effective in many cases, there are unlearning situations where it may not perform optimally. Future work could expand on our approach and generalize it to a broader range of unlearning algorithms, potentially addressing situations where gradient-ascent may fall short. Fourth, our method incorporates additional training objectives towards achieving future optimality, which may lead to a trade-off

in the model's current performance as well as extra computational overhead. Although this trade-off is not significant in our experiments, it could become more pronounced in larger-scale models and more complex deployment scenarios, where the data environment is dynamic and ever-evolving. Future studies could explore ways to mitigate this trade-off, such as identifying key factors that influence the balance between current model performance and future unlearning preparedness, and finding conditions under which such compromises can be minimized. Finally, while we focus on the MAML training paradigm in this study, we acknowledge that other meta-learning algorithms could also be useful in this context. Future research could explore more advanced meta-learning techniques to enable more efficient preparation, such as reducing the preparatory duration required to achieve meaningful unlearning readiness.

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

## A    ALGORITHM

---

**Algorithm 1** Ready2Unlearn: Learning with Unlearning Preparedness

---

1: **Input:** Initial model parameters $\theta_0$; training data $\mathcal{D}$, composed of forget data $\mathcal{D}_\mathrm{f}$ and retain data $\mathcal{D}_\mathrm{r}$; recovery data $\mathcal{D}_\mathrm{rc}$; number of outer-loop optimization steps $N$; adaptation rate $\alpha$;[10] learning rate $\eta$; loss weight coefficients $\lambda_1$, $\lambda_2$, and $\lambda_3$; and loss function $\mathcal{L}$.

2: **Output:** A model with unlearning preparedness $\theta_P$.

3: **for** $i = 1$ **to** $N$ **do**

4:    Sample $x_\mathrm{f} \sim \mathcal{D}_\mathrm{f}$, $x_\mathrm{r} \sim \mathcal{D}_\mathrm{r}$, $x_\mathrm{rc} \sim \mathcal{D}_\mathrm{rc}$

5:    $\hat{\theta}_{i-1} = \theta_{i-1} + \alpha \nabla_{\theta_{i-1}} \mathcal{L}\left(\theta_{i-1}; x_\mathrm{f}\right)$ *# Inner-loop update mimicking the unlearning.*

6:    $g_0 = \nabla_{\hat{\theta}_{i-1}} \mathcal{L}\left(\hat{\theta}_{i-1}; x_\mathrm{f}\right)$ *# For improving future unlearning efficiency.*

7:    $g_1 = \nabla_{\hat{\theta}_{i-1}} \mathcal{L}\left(\hat{\theta}_{i-1}; x_\mathrm{r}\right)$ *# To support capability retention after future unlearning.*

8:    $g_2 = \nabla_{\hat{\theta}_{i-1}} \mathcal{L}\left(\hat{\theta}_{i-1}; x_\mathrm{rc}\right)$ *# To enhance future post-unlearning resilience.*

9:    Sample $x \sim \mathcal{D}$

10:    $g_3 = \nabla_{\theta_{i-1}} \mathcal{L}\left(\theta_{i-1}; x\right)$ *# For maximizing current model utility.*

11:    Update $\theta_i \leftarrow \theta_{i-1} - \eta \left(-g_0 + \lambda_1 g_1 + \lambda_2 g_2 + \lambda_3 g_3\right)$ *# Outer-loop parameter update.*

12: **end for**

13: $\theta_P \leftarrow \theta_N$

14: **return** $\theta_P$

---

---

[10] Although the term "adaptation rate" does not refer to adaptation per se in our setting, we stick to it to maintain consistency with the meta-learning convention.

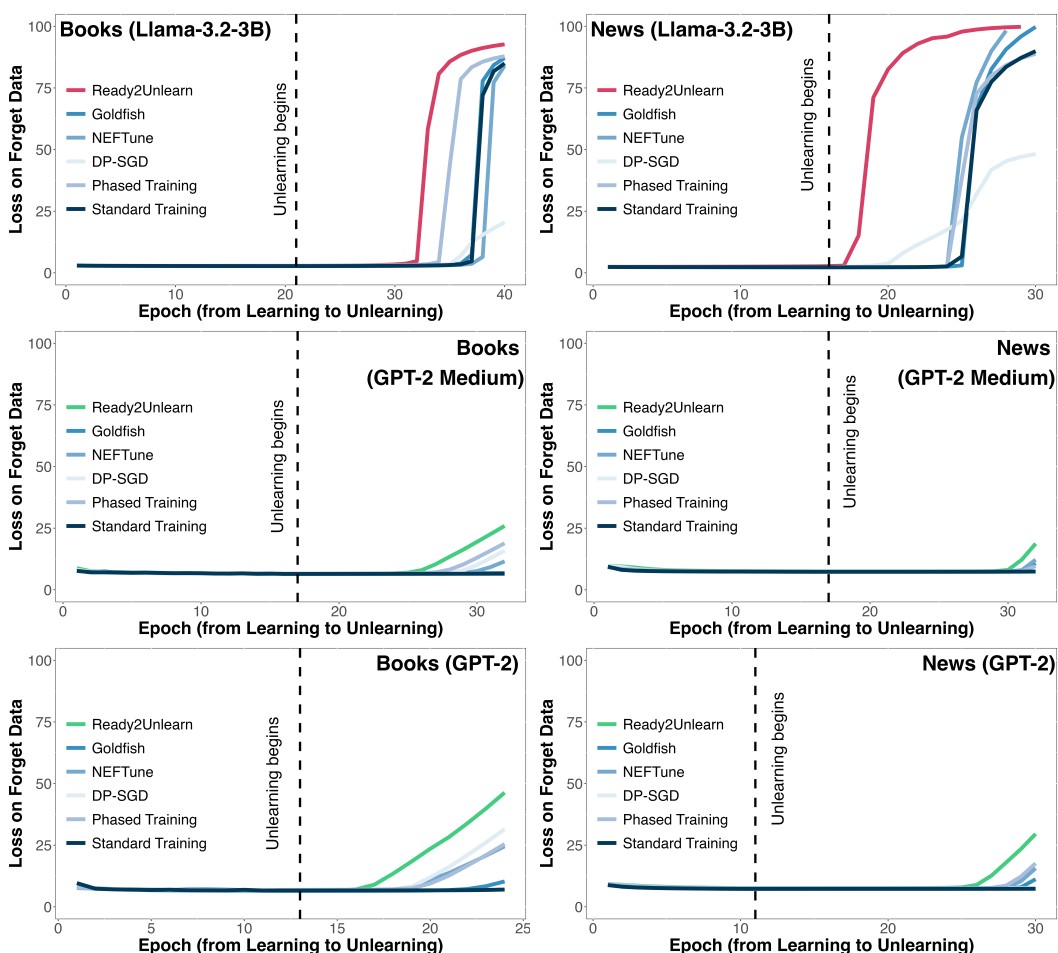

Figure 8: Extended comparison of unlearning efficiency in different language models: Llama-3.2-3B (top row), GPT-2 Medium (middle row), and GPT-2 (bottom row), on the MUSE-Books (left) and MUSE-News (right) datasets. The vertical axis represents the cross-entropy loss on the forget data, with each line corresponding to a different training-time strategy. During unlearning, we maintain a consistent unlearning rate of $1 \times 10^{-6}$ across all methods to ensure a fair comparison. The vertical dashed line marks the moment when unlearning begins. Significance tests confirm that, across all dataset-model combinations, at the Epoch when the forget loss of Ready2Unlearn first crosses 20, this loss is significantly larger than that achieved by the best competing baseline approach (all $p < 1 \times 10^{-7}$, one-tailed t-test, based on 10 independent trials).

## B  ADDITIONAL UNLEARNING EFFICIENCY EVALUATION IN MORE LANGUAGE MODELS

Please refer to Figure 8 for the evaluation results.

## C  IMPACT OF PREPARATION DURATION ON FUTURE UNLEARNING EFFICIENCY

We investigate how the duration of preparatory training with Ready2Unlearn influences the efficiency of subsequent unlearning. Specifically, we explore how varying the number of epochs dedicated to preparation during training affects the model's ability to forget target data efficiently when unlearning is later initiated. We consider a total training budget of 20 epochs. For each experimental setting, we allocate the final $M$ epochs to preparatory training using Ready2Unlearn, while the

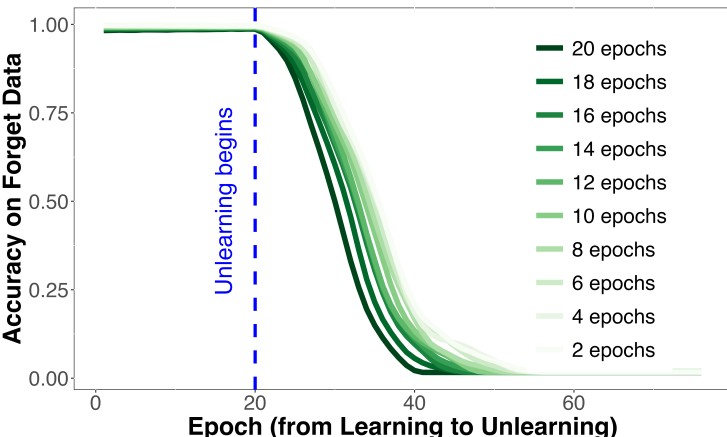

Figure 9: Effect of preparation duration on future unlearning efficiency with Ready2Unlearn. This figure visualizes the results for 10 distinct settings, where the number of epochs dedicated to preparation ranges from 2 to 20. The horizontal axis represents the timeline from the start of training to the completion of unlearning. The vertical axis shows the accuracy on the forget set, with lower values indicating more effective forgetting. The dashed line marks the epoch at which unlearning begins (epoch 20). All results are based on experiments conducted using the MNIST dataset.

preceding $(20 - M)$ epochs follow Standard Training procedures. For example, the setting labeled "6 epochs" represents a scenario in which the model undergoes 14 epochs of Standard Training, followed by 6 epochs of preparation using Ready2Unlearn. The "20 epochs" setting then corresponds to applying Ready2Unlearn throughout the entire training period. The results for 10 distinct settings ($M \in \{2, 4, 6, \ldots, 20\}$), presented in Figure 9, reveal a clear trend that longer preparation with Ready2Unlearn consistently leads to a faster response to future unlearning requests, with the model's performance on the forget data dropping earlier. This suggests that more extensive integration of Ready2Unlearn during training equips the model with stronger unlearning readiness.

## D    COMPUTATIONAL COST ANALYSIS

Consider that Ready2Unlearn optimizes additional meta-objectives during training, which may introduce extra runtime overhead. Here, we intuitively gauge such incremental computational cost by comparing the training runtimes of Ready2Unlearn and Standard Training. We define runtime as the duration required for the model to reach converged task performance, and report the relative training time overhead in Figure 10, specified as:

$$\text{Relative Overhead (\%)} = \frac{T_{\text{R2U}} - T_{\text{STD}}}{T_{\text{STD}}} \times 100, \tag{2}$$

where $T_{\text{R2U}}$ and $T_{\text{STD}}$ represent the training times for Ready2Unlearn and Standard Training, respectively.

The comparison reveals several notable observations. First, as expected, optimizing the extra meta-learning objectives results in increased training time. On average, we observe a 13.7% runtime overhead across model-task configurations. Second, this overhead varies with model architecture, task type, and data modality. Interestingly, smaller models (e.g., CNNs) generally experience a greater relative overhead in comparison to larger ones.[11] Third, we find that the variance in runtime across independent runs tends to be larger for high-capacity models (e.g., Llama-3.2) than for smaller ones (e.g., CNNs). This suggests that extremely large-scale models may be susceptible to unstable optimization dynamics under the added meta-objectives. We highlight this potential risk for practitioners and encourage future work to explore solutions that mitigate such variability and improve stability in large-scale training settings.

---

[11]This observation seems counterintuitive; we believe a more rigorous and comprehensive assessment that rules out potential confounding factors (such as task complexity) is warranted, and we leave this to future work.

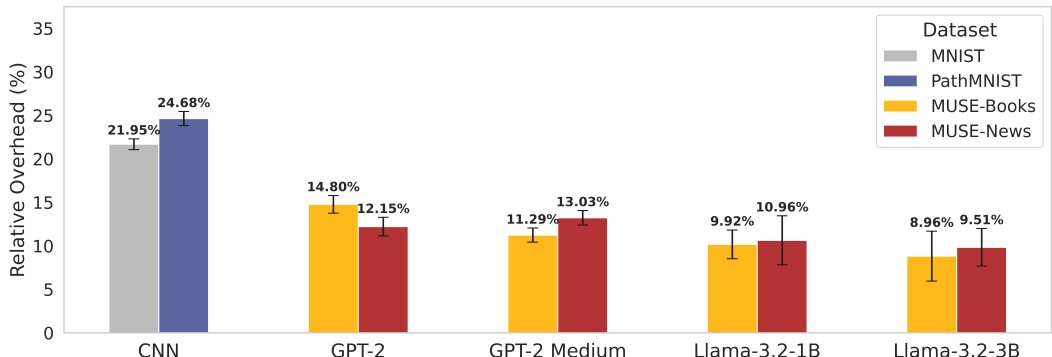

Figure 10: Relative training time overhead of Ready2Unlearn compared to Standard Training across different model-task configurations. Each configuration is evaluated over 10 independent trials.

Overall, there is no free lunch in preparing models in advance for proactive unlearning; nevertheless, we believe that investing a modest amount of extra training time is justified to hedge against potential future uncertainty.

## E   ABLATIONS

**Varying the retention loss weight** $\lambda_1$**.** We investigate the robustness of Ready2Unlearn by varying the retention loss scaling factor $\lambda_1$ from 1.0 to 4.0 (Table 1). When setting $\lambda_1 = 4.0$, once unlearning is triggered, Ready2Unlearn enables the model to maintain strong performance on the retain data, preserving over 80% of its pre-unlearning accuracy. Meanwhile, the accuracy on the forget data drops to 67.9%, indicating a moderate unlearning efficiency, with the model exhibiting a roughly 30% reduction in its ability to correctly recognize examples from the forget set. Further reducing $\lambda_1$ to 1.0 leads to a significant improvement in unlearning efficiency, with the relative drop in forget-set accuracy (compared to pre-unlearning performance) exceeding 40%, while the model still maintains a considerable portion of its pre-unlearning performance on the retain data (77.6%). It should be noted that while the pre-unlearning performance varies with different choices of loss weights, the degree of variation remains modest provided the weighting factors are chosen within a reasonable range. Overall, these observations demonstrate that $\lambda_1$ is indeed responsible for governing the trade-off between unlearning efficiency and retention performance, and can be adjusted to appropriately balance the two.

| Loss Weighting $\lambda_1$ | Pre-Unlearning Accuracy (%) | | Post-Unlearning Accuracy (%) | | Retain Rate (↑) | Forget Rate (↑) |
|---|---|---|---|---|---|---|
| | Retain (↑) | Forget (↑) | Retain (↑) | Forget (↓) | | |
| $\lambda_1 = 1.0$ | 96.4 | 94.2 | 74.8 | 54.2 | 77.6% | 42.5% |
| $\lambda_1 = 2.0$ | 98.6 | 97.8 | 75.6 | 56.7 | 76.7% | 42.0% |
| $\lambda_1 = 3.0$ | 98.2 | 98.6 | 78.6 | 65.4 | 80.0% | 33.7% |
| $\lambda_1 = 4.0$ | 97.4 | 98.6 | 80.5 | 67.9 | 82.6% | 31.1% |

Table 1: Pre-unlearning and post-unlearning classification performance on the retain and forget data when varying the retention loss weight $\lambda_1$. Results are based on the MNIST classification task described in the main text. The *retain rate* is computed as $\text{Acc}_{\text{post}}(\text{retain data})/\text{Acc}_{\text{pre}}(\text{retain data})$, and the *forget rate* is computed as $[\text{Acc}_{\text{pre}}(\text{forget data}) - \text{Acc}_{\text{post}}(\text{forget data})]/\text{Acc}_{\text{pre}}(\text{forget data})$. While varying $\lambda_1$, all other weights are held constant. Reported accuracies are taken from the model checkpoint at Epoch 30.

**Ablation results for the resilience term.** Here, we provide detailed results in Table 2, corresponding to Figure 6 in the main text. These ablation experiments isolate the effect of the resilience term in Equation 1 by comparing models trained with and without this term. The results demonstrate the role of the resilience term in preventing recovery of forgotten data during unlearning.

| Resilience Weighting $\lambda_2$ | Converged Loss on Forget Data | |
|---|---|---|
| | **Llama-3.2-1B** | **GPT-2** |
| $\lambda_2 = 3.0$ | 5.24 | 8.04 |
| $\lambda_2 = 0$ | 4.69 | 7.40 |

Table 2: Ablation results for the resilience term in Ready2Unlearn. The table shows the converged loss on forget-set data for Llama-3.2-1B and GPT-2 models with ($\lambda_2 = 3.0$) and without ($\lambda_2 = 0$) the resilience term. Higher loss indicates stronger resistance to unintentional relearning.

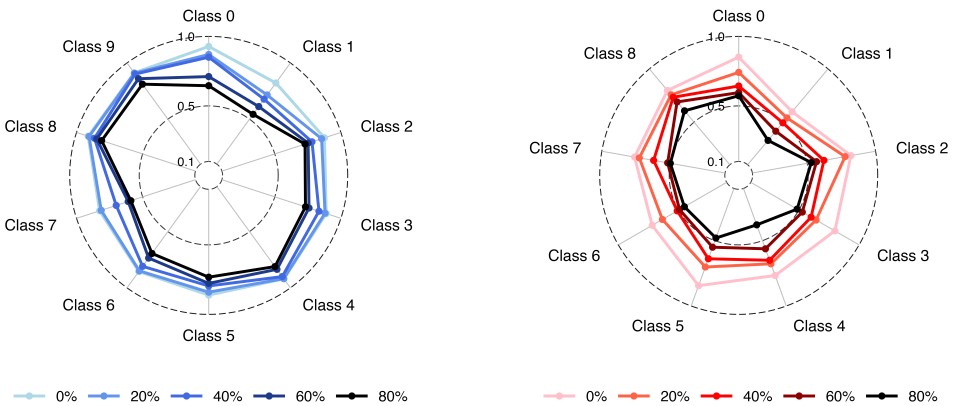

Figure 11: Retention performance under varying levels of overlap between forget and retain sets, shown for MNIST (left) and PathMNIST (right). The color gradient indicates the level of mixing, with darker shades corresponding to greater overlap; for example, "20%" means that 20% of the forget set is made up of misclassified retain samples.

## F  EFFECT OF FORGET-RETAIN OVERLAP

In practice, real-world forget and retain data may not be perfectly separable into the revocable and stable categories assumed by Ready2Unlearn, resulting in potential overlap between the two. In this section, we investigate this scenario using the MNIST and PathMNIST classification tasks (see setup described in the main text). We simulate overlaps by blending varying portions of the retain data into the forget set and report the corresponding retention performance in Figure 11.

We observe that as a larger portion of retain data is mistakenly included in the forget set, retention performance is slightly compromised after unlearning. This is expected, as the model allocates effort to prepare for "forgetting" data that ultimately should be retained, representing a suboptimal use of model capacity. Despite this, compared with the baseline results in Figure 4 of the main text, the model still performs better than if no proactive preparation had been undertaken at all. This suggests that proactive unlearning preparation is generally beneficial and that Ready2Unlearn tolerates reasonable levels of data miscategorization, demonstrating its practical utility.

Overall, this investigation indicates that while perfect categorization of forget and retain data may not be feasible in practice, Ready2Unlearn can still provide meaningful advantages over reactive or unprepared training approaches.

## G  VISUALIZING THE EFFECT OF THE RESILIENCE TERM

In the main text, from the generation-likelihood perspective (Figure 7), we show that the resilience term (the penultimate term in Equation 1) encourages the model to forget more distinctive, instance-specific information rather than generic features commonly shared across the broader data distribution. In this section, we provide additional supporting evidence for this effect through the lens of learned representations.

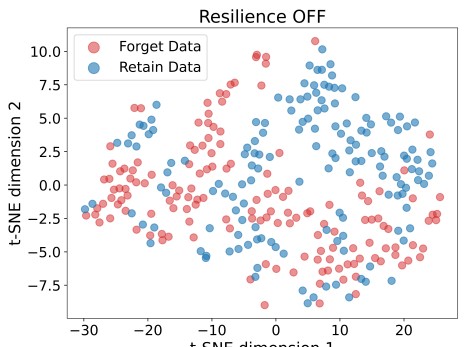 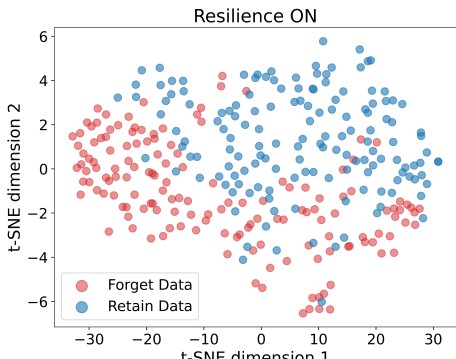

Figure 12: t-SNE visualization of forget and retain data representations for models trained with (right) and without (left) the resilience term. The results are based on GPT-2 and the Enron email dataset, as in the main text.

Specifically, we study two models: one trained with the resilience term active and one in which the resilience term is ablated from the optimization objective. We visualize the representations[12] of the forget and retain data produced by each model using t-SNE (Maaten & Hinton, 2008) in Figure 12.

Comparing the two subplots, for the model trained without the resilience term (left), the t-SNE embeddings of the forget and retain data exhibit substantial overlap, indicating that the model encodes these two groups along largely similar feature directions. In contrast, when the resilience term is included (right), the embeddings of the forget and retain data are much more separable, with noticeably less overlap. This increased separability suggests that the model has learned to encode the forget data using more distinctive, less broadly shared features, thereby making it harder for subsequent finetuning on data outside the forget set to regain the forgotten information.

Overall, this representational analysis provides additional empirical evidence for the "*distinctiveness-promoting*" role of the resilience term, complementing the token loss analysis in the main text. We hope this sheds light on developing appropriate loss functions to address key research challenges in targeted machine unlearning.

## H  SCALING PROPERTIES WITH RESPECT TO FORGET-DATA DIVERSITY

In practical scenarios, the forget set may consist of data that is highly diverse. In this section, we investigate how the performance of Ready2Unlearn scales with respect to the diversity of the forget data. Specifically, we extend the class-wise unlearning setup from the main text on MNIST. Instead of restricting the forget set to a single class, we gradually increase its diversity by allowing multiple classes to constitute the forget set; a larger number of included classes corresponds to a higher level of diversity. We report in Figure 13 the post-unlearning retention performance of models trained with Ready2Unlearn under varying levels of forget-data diversity, holding all other experimental settings fixed.

The results show that the diversity of the forget set has a statistically significant effect on the model's retention capability during unlearning. In general, retention performance decreases as forget-data diversity increases, since preparing the model to forget a broader range of features competes with its ability to preserve performance on the retained data under a fixed model capacity.

However, we also observe that this decline plateaus as diversity continues to increase, suggesting that the trade-off is bounded rather than unbounded. In other words, while more diverse forget data compromise retention during unlearning, the magnitude of the compromise stabilizes. We further expect that higher-capacity models may tolerate greater levels of forget-data diversity before notable degradation appears, a direction that we believe warrants more thorough future investigation.

---

[12]By "representation," we refer to the aggregated last-layer token hidden states obtained via average pooling.

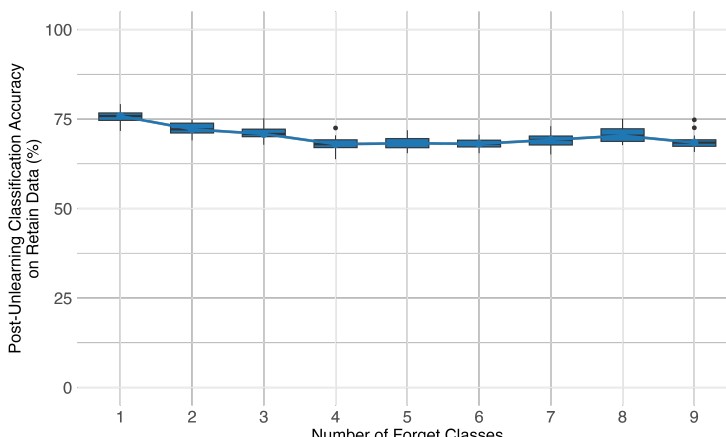

Figure 13: Post-unlearning retention performance for Ready2Unlearn-assisted training across varying levels of forget-data diversity. Results are based on the MNIST class-wise unlearning setup described in the main text, with retention performance averaged over classes. Each experiment is repeated over 10 independent trials.

## I  WHEN TO INTERVENE IN UNLEARNING PREPARATION

Given that Ready2Unlearn modifies the model parameters to a state optimized for future unlearning, it is possible that this "prepared" state can be partially altered or degraded by subsequent training on new tasks or additional epochs. In other words, the benefits of preparation may be compromised if further training shifts the model away from the carefully structured parameter configuration intended for efficient unlearning.

In this section, we investigate how the timing of the Ready2Unlearn intervention affects later unlearning efficiency. We adopt the MNIST digit unlearning setup described in the main text, where the overall training duration comprises 20 epochs. To study the effect of timing, we apply 4 epochs of Ready2Unlearn optimization at different stages of the training process: during the first quarter of training (epochs 1-5; Q1), the second quarter (epochs 6-10; Q2), the third quarter (epochs 11-15; Q3), and the final quarter (epochs 16-20; Q4). In all remaining epochs, standard training is applied. This design simulates varying intervention timings, with earlier interventions exposing the model to a higher likelihood of the prepared state being modified by subsequent training, while later interventions preserve the prepared state closer to the point of unlearning.

We evaluate unlearning efficiency across these four scenarios and compare them to a baseline in which no unlearning preparation is applied, i.e., standard training throughout. The results, presented in Figure 14, indicate that later interventions generally yield better unlearning performance. This is expected, as applying Ready2Unlearn closer to the end of training ensures that the prepared state remains largely intact when unlearning occurs. At the same time, these results highlight a potential risk: the prepared model state can be overwritten or partially degraded by other training updates, which should be noted as a caveat. Importantly, compared to the baseline without any preparation, all intervention timings provide measurable benefits. This suggests that performing unlearning preparation is almost always advantageous, regardless of timing, although the effectiveness is maximized when applied later in the training process.

Overall, this investigation sheds light on a subtle but important aspect of Ready2Unlearn: the timing of intervention matters, and careful consideration may help maximize the efficiency and robustness of later unlearning.

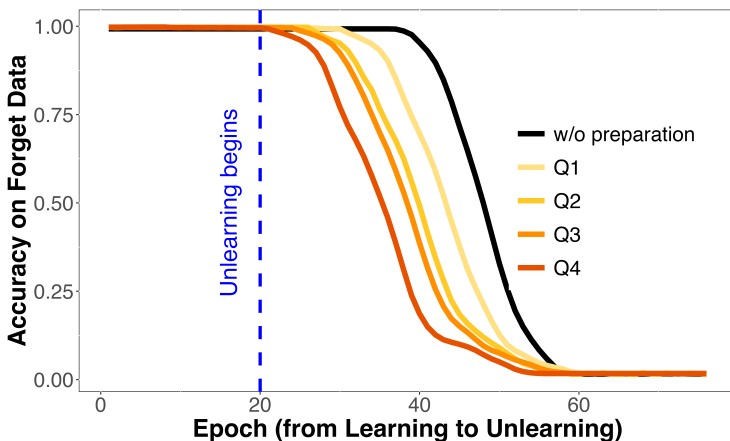

Figure 14: Effect of Ready2Unlearn intervention timing on unlearning efficiency. The intervention is applied during different quarters of the 20-epoch training process. The results are based on the MNIST task.

## J   EFFECT OF SIMILARITY BETWEEN FORGET AND RECOVERY DATA ON UNINTENTIONAL RELEARNING

Ready2Unlearn uses a recovery set to prevent unintentional relearning of forgotten data. An interesting question is how the similarity between the recovery and forget data influences this effect. In this section, we conduct additional experiments to investigate the impact of recovery-forget data similarity on the model's resistance to unintentional relearning.

To quantify similarity, we first encode each sample in both the forget and recovery sets into vectors using the text embedding model *gte-Qwen2-7B-instruct* (Li et al., 2023b). For each recovery sample, we compute its cosine similarity with every sample in the forget set and take the average similarity as a measure of how close that recovery sample is to the forget set. Based on these average similarities, we divide the recovery set into four quantiles: Q1 contains recovery samples least similar to the forget data, and Q4 contains the most similar ones.

We then perform Ready2Unlearn using each quantile of recovery data separately and report the post-recovery loss on the forget data in Table 3. Differences in converged loss across quantiles indicate that recovery-forget similarity plays an important role in the resulting model's resistance to unintentional relearning. In general, higher similarity between the recovery and forget data corresponds to greater resistance, as reflected by higher loss values observed when using the most similar quantile (Q4). Overall, we hope this analysis helps shed light on strategies for selecting efficient recovery data for more robust unlearning.

| Quantile by Similarity | GPT-2 | Llama-3.2-1B |
|:---:|:---:|:---:|
| 1 | 7.7 | 4.8 |
| 2 | 7.8 | 5.1 |
| 3 | 8.3 | 5.4 |
| 4 | 8.5 | 5.6 |

Table 3: Post-recovery loss on the forget data across four quantiles of recovery samples, divided by their average similarity to the forget set (Q1 = least similar, Q4 = most similar). Results are reported for GPT-2 and Llama-3.2-1B models, with higher loss indicating stronger resistance to unintentional relearning.

