# OpenReview forum: "Ready2Unlearn: A Learning-Time Approach for Preparing Models with Future Unlearning Readiness"
_ICLR.cc/2026/Conference — Submitted to ICLR 2026_

### Official Review · Reviewer_kNNt · 2025-10-28

**Soundness:** 1
**Presentation:** 2
**Contribution:** 3
**Rating:** 2
**Confidence:** 4

**Summary:**

The paper introduces Ready2Unlearn, a meta-learning inspired training pipeline that prepare the a particular deep machine learning model to be able to easily unlearn. Ready2Unlearn uses meta-learning principles to train models in a way that makes them easier to unlearn later. Specifically, it simulates future unlearning via gradient ascent during pre-training, and optimizes the model to (1) forget efficiently, (2) retain useful knowledge, and (3) counter relearning forgotten data. Based on the formulation provided the method is model-agnostic. Based on the experiments conducted on image and language tasks show that Ready2Unlearn improves unlearning speed, robustness, and performance retention compared to standard and baseline approaches.

**Strengths:**

1. The idea of making models unlearning-ready at training time is a significant conceptual shift from existing reactive methods.

2. Proactive unlearning via meta-objectives is a novel contribution in the unlearning space. While some prior works use meta-learning for unlearning itself, this is the first to integrate it during the initial training phase.

3. Partitioning training data into revocable (likely to be unlearned) and stable categories and incorporating this into the meta-objective is novel, despite its overly-simplified assumption.

**Weaknesses:**

1. While the results are averaged, there are no confidence intervals or statistical significance tests, limiting interpretability and reliability, especially in sensitive areas like privacy.

2. The added cost of meta-learning during training (e.g., outer-loop gradients, more iterations) is not thoroughly quantified or compared. For example, meta learning would be hard to scale to large-scale models, and given the context-dependent and evolving nature of high-risk and low-risk splits it may be hard to adjust the model dynamically at the time of unlearn-aware training.

3. The framework assumes training data can be split neatly into revocable and stable categories, which may not hold in practice. In reality, these categories classification may be context-dependent, evolving, or uncertain. Hence, it seems that making the model to be unlearning aware is a bit of an overcommitment. In practice, knowing what data may be subject to future unlearning isn't always feasible, making the deployment of the framework non-trivial in many real-world applications.

**Questions:**

1. How does Ready2Unlearn perform when the forget set overlaps significantly with the retain set?
2. Can the meta-objective be extended to support non-gradient ascent based unlearning techniques?
3. Is it feasible to adapt Ready2Unlearn in settings where the forget set is unknown at training time?
4. How sensitive is the performance to $\lambda_1$, $\lambda_2$, $\lambda_3$ loss weights, and are there principled ways to set them?
5. Can this approach scale to very large models and datasets without significant increases in training cost?

---

> ### Author Response · Authors · 2025-11-30
>
> Summary:
>
> **Response:** The authors sincerely appreciate the reviewer’s careful consideration and the effort invested in reviewing the manuscript. The reviewer’s summary perfectly distills the essence of the work, which the authors greatly appreciate. All of the reviewer’s comments are fully respected, and a point-by-point response is provided below. The authors look forward to working collaboratively with the reviewer to further strengthen the manuscript through this constructive exchange.
>
> S1:
>
> **Response:** The authors are pleased to see that the reviewer recognizes this paradigm shift.
>
> S2:
>
> **Response:** The authors thank the reviewer for acknowledging the idea of “proactive unlearning” as a novel contribution to the literature. The authors are pleased that this central message they wish to convey to the community is accurately captured.
>
> S3:
>
> **Response:** Thank you to the reviewer for acknowledging the novelty of formulating the meta-objective with revocable/stable data categorization. The authors greatly appreciate this recognition. Regarding the reviewer’s concern about the “overly-simplified assumption,” the authors respect this comment and address it in detail in a subsequent response to a related comment.
>
> W1:
>
> **Response:** The authors would like to thank the reviewer for bringing this rigor issue to their attention and acknowledge this oversight in the initial submission. In the revised manuscript, the authors have updated the captions of all benchmarking figures, where relevant statistical test results are now provided. The reason for choosing to report the significance test results rather than show confidence intervals is primarily to maintain visual clarity, and the authors hope the reviewer finds this approach acceptable. The authors kindly invite the reviewer to refer to the updated figure captions for these additions. Again, the authors thank the reviewer for this valuable comment.
>
> W2:
>
> **Response:** Thank you to the reviewer for this insightful comment. The authors acknowledge that, in practice, the implementation of Ready2Unlearn may face more complex industrial scenarios than the simplified testbed presented in the manuscript. The reviewer’s observations, including potential challenges in scaling to large models and managing a dynamic data environment, are all valid points that warrant consideration for practical deployment.
>
> That said, the authors kindly invite the reviewer to also consider this study from a broader perspective. The primary goal of this work is to introduce and motivate a proactive direction within the unlearning community and to provide a conceptual stepping stone in a simplified environment, serving as a “food for thought” to encourage further discussion on proactive and forward-looking approaches in this area.
>
> Nonetheless, the authors take the reviewer’s concerns seriously and, in response, have **added a brief discussion** of potential practical challenges in **lines 539 to 546 on page 10**, and **included a new cost analysis in Appendix D on page 17** of the updated manuscript to give a clearer sense of the additional computational overhead introduced by the meta-learning objectives, as suggested by the reviewer. The authors hope that these additions satisfactorily address the reviewer’s comments.

---

> ### Author Response · Authors · 2025-11-30
>
> W3:
>
> **Response:** The authors appreciate the reviewer’s insightful comment concerning the realism of the revocable/stable data categorization. The authors acknowledge that, in practice, model developers or organizations (as data controllers) often cannot perfectly forecast at training time which data subjects may later request revocation of their associated data, as they naturally do not have oracle-like foresight regarding future unlearning requests.
>
> That said, the authors would like to respectfully share an example with the reviewer, noting that in many real-world enterprise settings, organizations often possess substantial domain knowledge and operational experience that can naturally inform and justify such categorizations.
>
> For example, in institutional practices under GDPR, as documented in BankingHub (**https://www.bankinghub.eu/finance-risk/gdpr-deep-dive-implement-right-forgotten?utm_source=chatgpt.com**), GDPR requires “institutions to achieve a much deeper understanding of the purpose for which personal data is kept. In order to do this, each item of information will need to be classified, not only by its purpose but also by the source from which it has been collected.”
>
> Here, it is observed that institutions routinely classify customer data to comply with GDPR requirements, considering both its purpose (e.g., marketing, initiation of business) and, more crucially, its source (e.g., publicly accessible, provided directly by the individual, derived from the institution’s systems, or inferred/behavioral data) (see Figure 3 in the article for details).
>
> The authors believe that such routinely conducted data analytics practices in organizations, particularly the categorization based on the source from which data is collected, can serve as strong indicators of unlearning risk (e.g., data from individuals associated with a higher likelihood of unlearning requests compared to those publicly accessible), thereby providing a natural environment for practitioners to adopt proactive strategies with Ready2Unlearn. The authors hope this example helps to “beef up” the practical relevance of the proposed framework. That said, the authors take the reviewer’s comment seriously and have **explicitly highlighted this point** in the updated manuscript **(lines 94–101, page 2)**.
>
> Additionally, to **further justify this setup**, we conduct **new experiments in Appendix F**. These results demonstrate that, although perfect categorization of forget and retain data is rarely achievable in practice, Ready2Unlearn can tolerate a reasonable degree of miscategorization and still provide meaningful advantages over reactive or unprepared training approaches. We kindly invite the reviewer to review this new content.
>
> Q1:
>
> **Response:** The authors thank the reviewer for raising this practically relevant question and agree that overlap between the two sets is indeed possible in practice and is an important scenario to evaluate.
>
> To address this, the authors have **added new experiments in Appendix F (page 19)**, where varying levels of overlap between the forget and retain sets are simulated. Overall, the results indicate that retention performance decreases as the level of overlap increases, although the performance drop remains minor. This behavior is expected, given the inherently conflicting objectives of forgetting and retaining.
>
> The authors thank the reviewer for prompting this interesting experiment and kindly invite the reviewer to refer to Appendix F (page 18) of the revised manuscript for details and hope that these additions satisfactorily address the reviewer’s question.
>
> Q2:
>
> **Response:** Brilliant point! While the current formulation of Ready2Unlearn uses gradient ascent-based meta-objectives, the underlying principles are not inherently tied to gradient ascent. In principle, the meta-objective can be extended to support alternative unlearning techniques by appropriately defining the unlearning simulation in the meta-training loop.
>
> The authors believe this is a promising direction and, in response, **have added a discussion** of this potential **in lines 535–539 on page 10** of the revised manuscript. They hope this addition may inspire creative follow-up work in this direction and would like to thank the reviewer for raising this interesting and practically relevant implication.

---

> ### Author Response · Authors · 2025-11-30
>
> Q3:
>
> **Response:** Thank you, the reviewer, for this practically relevant question. Indeed, it is challenging to prepare the model in advance if nothing is known about potential future unlearning. That said, the reviewer’s comment motivates potential extensions of Ready2Unlearn, such as incorporating uncertainty estimation into the design of the meta-objective or using probabilistic approaches to anticipate likely unlearning cases. The authors appreciate the reviewer’s valuable insights and **have added a discussion** of these possibilities **in lines 527–532 on page 10** of the revised manuscript. The authors thank the reviewer for prompting this meaningful discussion.
>
> Q4:
>
> **Response:** The authors thank the reviewer for asking for this clarification on hyperparameter choice. To answer the reviewer’s question, throughout all experiments in this work, the loss weight parameters were selected based on a held-out validation set, following the principle that the chosen weights should enable the fastest unlearning while ensuring that the training performance does not drop by more than 1% relative to the same model trained without the meta-objectives. The authors acknowledge the oversight in not providing this information in their initial submission and have updated the manuscript to **include these details in lines 345–346 on page 7**.
>
> Additionally, the authors **have added experiments** to evaluate the robustness of Ready2Unlearn under varying loss weights, with the results provided **in Appendix E on page 18** of the revised manuscript.
>
> The authors kindly invite the reviewer to review the new content and hope that these additions satisfactorily address the reviewer’s question.
>
> Q5:
>
> **Response:** Thank you, the reviewer, for raising this important question. This is indeed a practical point, as the inclusion of additional meta-objectives inevitably introduces extra training cost, particularly for larger-scale models. The authors take this comment seriously and, in response, **have added a training cost analysis in Appendix D on page 17** of the revised manuscript.
>
> The authors hope that this addition provides readers with a clearer sense of the potential computational overhead introduced by the meta-learning objectives and that it satisfactorily addresses the reviewer’s question.
>
> The authors sincerely thank the reviewer for the helpful comments, which have greatly contributed to strengthening this work. They would be grateful if the reviewer could kindly re-evaluate the manuscript in light of the revisions. Should the reviewer have any further questions, the authors would be pleased to address them. Thanks.
>
> -- Authors of Submission 15075

---

### Official Review · Reviewer_heUx · 2025-10-30

**Soundness:** 3
**Presentation:** 4
**Contribution:** 4
**Rating:** 8
**Confidence:** 4

**Summary:**

This paper introduces Ready2Unlearn, a meta-learning framework that prepares models to support future unlearning efficiently and reliably. Instead of treating unlearning as a post-hoc reactive process, Ready2Unlearn applies meta-learning optimization during training to anticipate future unlearning requests. The method explicitly optimizes three “forward-looking” objectives, efficiency, retention, and resistance, so that later gradient-ascent-based unlearning is faster, preserves overall performance, and resists recovery of forgotten data. Experiments across image classification (MNIST, PathMNIST) and language tasks (MUSE-Books, MUSE-News, LLaMA/GPT-2) show improved unlearning speed and robustness compared to several baselines (loss reweighting, DP-SGD, NEFTune, etc.).

**Strengths:**

1. Novel Research Direction: The paper introduces a genuinely new direction for the machine unlearning community by emphasizing preparation for unlearning during the training stage. This proactive framing, rather than the usual reactive approach, is both novel and conceptually interesting.
2. Writing Clarity:  The writing is clear and well-organized. The argument flows naturally, and Figure 2 does an excellent job of visually illustrating the key idea. The authors make good use of space by focusing the main text on intuitive explanations and leaving the algorithmic details to the appendix.
3. Method Simplicity: The method itself is simple and practical. It is well motivated, clearly explained with minimal notation, and appears easy to implement across different architectures and codebases.

**Weaknesses:**

1. Time/Memory Complexity: While the motivation behind the approach is clear and compelling, the paper does not discuss the additional computational or memory costs introduced by the meta-learning framework. Information about training time, memory footprint, and convergence behavior would be valuable for assessing the practical feasibility of the method, especially for large-scale models.
2. Forget Size Limit: The paper does not clearly address the scalability of the proposed approach with respect to the size or diversity of the forget set. It would be helpful to understand how computational requirements and unlearning performance scale as the number or diversity of forget samples increases.
3. Unintended Consequences: While the paper does not report any unintended side effects, it would be helpful for the authors to discuss potential consequences of the proposed training structure. For example, could a model trained with unlearning preparedness become less adaptable when learning new tasks, as in standard continual learning scenarios? Additionally, if the model is later fine-tuned on many different tasks or asked to unlearn data outside the original forget set, might this shift it away from the “prepared” loss basin and diminish its unlearning efficiency, retention, or robustness?
4. Relearning Set Definition: The paper describes the relearning set as data with “similar distributional characteristics, such as stylistically or semantically similar examples,” but this notion remains somewhat vague. It would help if the authors provided a more formal definition or concrete examples of what constitutes such similarity. For instance, how similar can the relearning set be to the forget set before it no longer serves as a valid proxy for recovery evaluation? The boundary between the two is unclear, particularly since the relearning experiment is presented for text generation but not for image classification. Would fine-tuning on digits not included in the forget class, for example, still fall within the intended definition of relearning?
5. Experimental Clarifications: In Figure 6, the difference between the plateau points for the prepared and unprepared models appears relatively small. A more detailed quantitative analysis, perhaps through an ablation study, could clarify how each component of the final objective, especially the term intended to enhance resistance to relearning, contributes to this effect.

**Questions:**

1. The resilience term is motivated as a way to direct the model toward learning more distinctive, data-specific features of the forget set, thereby preventing easy relearning. Do you observe this effect empirically, for example, through feature visualizations showing that the inclusion of the resilience term makes the forget and retain representations more disjoint?

---

> ### Author Response · Authors · 2025-11-30
>
> Summary:
>
> **Response:** The authors sincerely appreciate the reviewer’s careful consideration and the time invested in evaluating the manuscript. The reviewer’s summary perfectly captures the key idea the authors aim to convey in this paper, which the authors greatly appreciate. The authors fully respect all of the reviewer’s comments, and a detailed point-by-point response is provided below. Once again, the authors thank the reviewer for the overall positive evaluation of the manuscript and look forward to working together with the reviewer to further strengthen this work through constructive exchange.
>
> S1:
>
> **Response:** The authors appreciate the reviewer’s recognition of the proactive paradigm introduced in this work and consider it novel and interesting. The authors are pleased that this central message, which they wish to convey to the unlearning community through this paper, has been accurately identified by the reviewer. Thanks!
>
> S2:
>
> **Response:** The authors would like to thank the reviewer for the encouraging words regarding the presentation of the work and are pleased that the reviewer finds the explanation intuitive.
>
> S3:
>
> **Response:** Thank you to the reviewer for considering the proposed method well-motivated. The authors are delighted that the reviewer finds the method clearly explained and easy to implement.
>
> W1:
>
> **Response:** The authors thank the reviewer for raising this important and practically meaningful point. The authors acknowledge that incorporating the additional meta-objectives during training introduces extra computational overhead. In response to the reviewer’s suggestion, the authors **have added a computational cost analysis in Appendix D (page 17)** of the revised manuscript. This new section provides discussion of the training-time computational overhead of the proposed method. The authors kindly invite the reviewer to review the newly added analysis, and hope that these additions offer a clearer sense of the computational cost of Ready2Unlearn for potential users, and hope that the reviewer finds this revision satisfactory.
>
> W2:
>
> **Response:** Brilliant point! The authors appreciate the reviewer for raising this insightful angle. This comment highlights an important yet practical consideration. In response, the authors **have added experiments** examining how the proposed method scales with respect to the diversity of the forget set, as suggested by the reviewer. The corresponding results and analysis are provided in **Appendix H (page 20)** of the updated manuscript. The authors kindly invite the reviewer to review this new content and hope that these additions are useful in providing a clearer sense of how the method scales with forget-set diversity. The authors hope the reviewer finds this revision satisfactory.
>
> W3:
>
> **Response:** Thank you, the reviewer, for noting this excellent point. The authors find this “unintended consequences” perspective particularly intriguing! The authors agree that exploring potential side effects is meaningful when introducing a new paradigm. In response, the authors **have added experiments** to examine possible side effects in “continual learning scenarios”, as suggested by the reviewer. The authors kindly invite the reviewer to refer to **Appendix I (page 21)** of the revised manuscript to review this newly added analysis. The authors hope that these additions contribute to the completeness of this work and hope the reviewer finds them satisfactory.
>
> W4:
>
> **Response:** The authors thank the reviewer for raising this important clarification. In response, the authors have **provided a clarification in lines 227–229 on page 5** of the updated manuscript. Furthermore, the reviewer’s question regarding “how similar can the relearning set be to the forget set before it no longer serves as a valid proxy for recovery evaluation” motivated the authors to extend the analysis. **New experiments** exploring varying levels of similarity between the relearning and forget sets **have been added in Appendix J (page 22)** of the revised manuscript. The authors kindly invite the reviewer to review this new content and hope that these additions help clarify the relearning scenario.
>
> Regarding the reviewer’s question on digit classification, the authors’ quick answer is “yes”, as fine-tuning on other digits, even those not included in the forget set, can still contribute to learning useful features for identifying the forget digit and therefore falls within the intended definition of relearning. The authors appreciate the reviewer for highlighting the previous vagueness in our relearning definition and hope the reviewer finds these revisions helpful in improving clarity.

---

> ### Author Response · Authors · 2025-11-30
>
> W5:
>
> **Response:** The authors thank the reviewer for bringing this important clarification to their attention. The suggestion to conduct an ablation study with quantitative analysis is valuable. In response, the authors **have added ablation experiments**, with results presented in **Appendix E (page 18)** of the updated manuscript. The authors kindly invite the reviewer to refer to this new content and hope these additions help improve clarity and satisfactorily address the reviewer’s comment.
>
> Q1:
>
> **Response:** Brilliant comment! The authors are pleased to see that the reviewer has accurately understood the motivation behind the resilience term. The suggestion to empirically visualize its effect is particularly insightful and valuable! In response, the authors **have conducted additional experiments** examining the impact of the resilience term, and have generated feature **visualizations illustrating its effect**. The results are provided in **Appendix G (page 19)** of the revised manuscript. The authors greatly appreciate this thoughtful suggestion and kindly invite the reviewer to review this new content. They hope that these additions offer further supporting evidence and satisfactorily address the reviewer’s comment.
>
> The authors sincerely thank the reviewer for the thoughtful and constructive feedback, which has substantially contributed to improving the quality of this work. Should any additional questions arise, the authors would be glad to address them. Thanks.
>
> -- Authors of Submission 15075

---

### Official Review · Reviewer_kqD5 · 2025-10-30

**Soundness:** 2
**Presentation:** 2
**Contribution:** 2
**Rating:** 2
**Confidence:** 3

**Summary:**

The authors propose Ready2Unlearn (R2U), which prepares models for future unlearning requests proactively during training. They use some MAML inspired dual-loop optimization to optimize for resistance metrics for when unlearning occurs later.

**Strengths:**

1. The novel perspective of shifting from reactive to proactive unlearning is interesting and paradigmaticallly different.
2. The method is model agnostic and doesn't require architecture specific unlearning algorithms.
3. The visual comparisons and systematic evals across many datasets provide clear evidence of effectiveness in the setup context.

**Weaknesses:**

1. The core assumption of the paper regarding being able to reliably predict which data is "revocable" vs "stable" is unrealistic, and the paper provides no principled approach for making this classification.
2. Due to a lack of theoretical foundation, it is hard to understand why this approach should work. There is no convergence analysis as well.
3. The proposed experimental setups where there are designated forget and retain classes is not realistic.
4. There is no baseline comparison against existing methods like influence-function based approaches or differential privacy techniques.

**Questions:**

1. How does the method handle cases where the revoke-stable categorization is wrong?
2. What happens when multiple overlapping unlearning requests occur?
3. What is the rationale behind using gradient ascent based unlearning when established parallel methods use other approaches?
4. How does performance change when recovery attacks use the actual forgotten data?

---

> ### Author Response · Authors · 2025-11-30
>
> Summary:
>
> **Response:** The authors sincerely appreciate the reviewer’s time and thoughtful assessment of the manuscript. All of the reviewer’s comments are fully respected, and a point-by-point response is provided below. The authors look forward to working together with the reviewer to strengthen the manuscript through this constructive exchange.
>
> S1:
>
> **Response:** The authors appreciate the reviewer’s recognition of the paradigm shift presented in the present work. The authors are pleased that the “from reactive to proactive” mindset, a central message we aim to convey to the community, has been accurately identified.
>
> S2:
>
> **Response:** The authors appreciate the reviewer’s embrace of the agnosticism in algorithm design.
>
> S3:
>
> **Response:** The authors appreciate the reviewer’s recognition of the method’s effectiveness.
>
> W1:
>
> **Response:** The authors sincerely thank the reviewer for this thoughtful comment. The authors would also like to respectfully share with the reviewer a scenario described below, noting that in many real-world enterprise settings organizations often possess substantial domain knowledge and operational experience that naturally inform and justify such categorizations.
>
> Following the above, the authors would like to share a real-world scenario based on institutional practices under GDPR. As documented in BankingHub (**https://www.bankinghub.eu/finance-risk/gdpr-deep-dive-implement-right-forgotten?utm_source=chatgpt.com**), GDPR requires “institutions to achieve a much deeper understanding of the purpose for which personal data is kept. In order to do this, each item of information will need to be classified, not only by its purpose but also by the source from which it has been collected.”
>
> Here, it is observed that institutions routinely classify customer data to comply with GDPR requirements, considering both its purpose (e.g., marketing, initiation of business) and, more crucially, its source (e.g., publicly accessible, provided directly by the individual, derived from the institution’s systems, or inferred/behavioral data) (see Figure 3 in the article for details).
>
> The authors believe that such routinely conducted data analytics practices in organizations, particularly categorization based on the source from which data is collected, can strongly predict unlearning risk (e.g., data from individuals associated with higher unlearning risk), providing a natural environment for practitioners to take proactive strategies with Ready2Unlearn.
>
> Additionally, to further justify this setup, we conduct **new experiments in Appendix F**. These results demonstrate that, although perfect categorization of forget and retain data is rarely achievable in practice, Ready2Unlearn can tolerate a reasonable degree of miscategorization and still provide meaningful advantages over reactive or unprepared training approaches. We kindly invite the reviewer to review this new content.
>
> Regarding proposing a principled categorization approach, the authors respectfully argue that they do not prescribe a specific method, primarily because the appropriate approach may vary substantially across different industrial contexts. For example, organizations often rely on their own experience and best practices to perform such categorizations in ways that align with their business needs. That said, the authors take note of the reviewer’s concern and have **added a brief clarification** in the updated manuscript providing high-level guidelines for this classification. The authors kindly invite the reviewer to refer to **lines 94–103 on page 2** of the updated manuscript for details.
>
> W2:
>
> **Response:** The authors thank the reviewer for the comment and would like to respectfully clarify this point. The introduced Ready2Unlearn approach is essentially grounded in the well-established MAML algorithm, a meta-learning paradigm with strong theoretical foundations. The authors believe that MAML provides a solid theoretical backbone for the proposed method. Additionally, the main contribution of this work does not lie in advancing the MAML algorithm itself; rather, it primarily contributes by introducing the meta-learning idea to the field of machine unlearning with a novel use case (i.e., shifting reactive unlearning to a proactive mindset). The authors take the reviewer’s comment seriously and have **highlighted this point in lines 188–191 on page 4** of the updated manuscript. The authors hope this clarification helps convey the main focus of the present study and satisfactorily addresses the reviewer’s concern.

---

> ### Author Response · Authors · 2025-11-30
>
> W3:
>
> **Response:** The authors thank the reviewer for noting this issue. This is indeed a valid point, and the authors acknowledge that explicitly designating forget and retain data is typically not the case in real-world scenarios, where data environments are often more complex and less structured.
>
> That said, the authors would like to respectfully clarify the overarching intent of this work. Rather than attempting to perfectly replicate real-world deployment settings, the primary goal of this study is to provide a conceptual stepping stone, a proof of possibility that brings attention to a forward-looking perspective within unlearning community. This shift from a reactive to a proactive paradigm has been consistently acknowledged by the review team, which the authors appreciate.
>
> From this standpoint, the authors believe that the current form of experimental setup is appropriate for demonstrating the feasibility and conceptual soundness of the proposed Ready2Unlearn framework in this novel use case. Moreover, assigning forget and retain classes is a common practice in the unlearning literature for evaluating and benchmarking new algorithms (Zhou et al. 2023).
>
> The authors fully understand the reviewer’s concern regarding realism, while they respectfully hope the reviewer may also consider this study from a broader perspective: this work aims to introduce and motivate a new proactive direction within the unlearning community, one that the authors hope will inspire further exciting work in the future. The authors appreciate the reviewer’s role in making this happen.
>
> W4:
>
> **Response:** Thanks to the reviewer for bringing up this important category of methods. The authors acknowledge that considering the influence of individual data can help mitigate the unlearning effort and **have included a baseline method**, termed DP-SGD, based on differential privacy techniques (Dwork 2006, Abadi et al. 2016), which limits the influence of individual data points on the model’s outputs, following prior work by Li et al. (2021). Details are provided on **page 9** of the updated manuscript, with corresponding results reported in **Figure 5 (page 9) and Figure 8 (see Appendix B, page 16)**. The authors hope the benchmarking protocol is satisfactory to the reviewer.
>
> Q1:
>
> **Response:** The authors thank the reviewer for raising this excellent question. In practice, it is indeed possible that model developers or organizations (as data controllers) cannot perfectly forecast (at model training time) which data subjects will later request revocation of their associated data after model deployment, as they naturally do not have oracle-like foresight regarding future unlearning requests. To be honest, the proposed method does not provide incremental benefits for data whose revocation status was incorrectly forecasted; such misjudged instances would be handled in the same manner as in standard reactive unlearning methods.
>
> The authors appreciate the reviewer’s thoughtful comment and, in response, have **added a discussion of this boundary condition** to the updated manuscript **(see lines 531–536, page 10)** to clarify the method’s limitations under potential misjudgment of revocable versus stable data. The authors again thank the reviewer for this remark, which helps to make the overall scope and applicability of the method clearer.
>
> Q2:
>
> **Response:** The authors thank the reviewer for raising this thoughtful and practical question. If understood correctly, by “overlapping unlearning requests” the reviewer is referring to scenarios in which multiple data subjects submit revocation requests in close succession, such that a second request arrives while the system is still processing a previous one.
>
> If this understanding is consistent with what the reviewer had in mind, the authors would like to clarify that although Ready2Unlearn is not specifically designed to manage concurrent unlearning operations, it still provides meaningful benefits in such situations. One of the primary goals of Ready2Unlearn is to reduce the duration and computational cost of each individual unlearning process, which is reflected in the Efficiency objective (the first term in Equation 1, defined in lines 218–222 on page 5). By shortening the time required to complete a single unlearning task, Ready2Unlearn indirectly helps systems manage overlapping requests more effectively, reducing the backlog of pending requests and shortening the waiting time for any request that arrives while another is being processed. The authors hope the above explanation satisfactorily addresses the reviewer’s question.

---

> ### Author Response · Authors · 2025-11-30
>
> Q3:
>
> **Response:** The authors thank the reviewer for this important question. The reason for choosing gradient ascent–based unlearning is that Ready2Unlearn proactively guides the model parameters during training toward a state that is particularly amenable to subsequent gradient ascent–based parameter updates. In other words, the model is “pre-conditioned” to respond efficiently when gradient ascent is later applied to remove specific data, making unlearning faster and more effective. In response, the authors **have added a discussion** on this point in **lines 113–117 on page 3** of the updated manuscript. The authors again appreciate the reviewer’s question and hope this helps clarify the rationale behind the choice of unlearning method.
>
> Q4:
>
> **Response:** The authors appreciate the reviewer bringing up this point. Essentially, if the exact forgotten data were used again to further fine-tune the model, the model would almost certainly recall that data. That said, the authors respectfully note that this scenario is uncommon in real-world settings. Typically, data controllers, such as organizations, delete the requested data from their databases once a data subject requests removal. This practice prevents the exact forgotten data from being reused in training, which is why in this work only similar data (representing inadvertent recovery) is used for testing. The authors appreciate the reviewer’s question and hope that the above explanation will help clarify the real-world relevance considered in the experimental design, and that the reviewer finds it reasonable.
>
> The authors thank the reviewer for the insightful feedback, which has been particularly helpful in strengthening this work. The authors would be grateful if the reviewer could kindly re-evaluate the work in light of the revised manuscript. Should the reviewer have any further questions, the authors would be happy to address them. Thanks.
>
> -- Authors of Submission 15075
>
> **References:**
>
> Abadi M, Chu A, Goodfellow I, McMahan HB, Mironov I, Talwar K, Zhang L (2016) Deep learning with differential privacy. Proceedings of the 2016 ACM SIGSAC conference on computer and communications security 308–318.
>
> Dwork C (2006) Differential privacy. International colloquium on automata, languages, and programming (Springer), 1–12.
>
> Li X, Tramer F, Liang P, Hashimoto T (2021) Large language models can be strong differentially private learners. arXiv preprint arXiv:2110.05679.
>
> Zhou J, Li H, Liao X, Zhang B, He W, Li Z, Zhou L, Gao X (2023) A unified method to revoke the private data of patients in intelligent healthcare with audit to forget. Nature Communications 14(1):6255.

---

### Meta-Review · Area_Chair_7kCm · 2026-01-04

**Summary:**

The paper proposes a meta-learning approach for preparing a model for future unlearning during training time. This represents a shift from addressing unlearning "reactively" (which most but not all prior unlearning methods do) to addressing unlearning needs "proactively" via a dedicated training phase.
More concretely, the authors propose a method called Ready2Unlearn that uses a MAML-like training objective that directly optimizes the model for speed of future unlearning requests (for the gradient ascent unlearning algorithm), retention of permissible knowledge, and resistance to future attempts to recover the forgotten knowledge.

The main concerns of the reviewers are:
- **C1**. Unrealistic assumption that we can predict which data is “stable” vs “revocable” ahead of time (Reviewer kqD5, Reviewer kNNt)
- **C2**. No theoretical foundation or convergence analysis (Reviewer kqD5)
- **C3**. Unrealistic experimental setup (with designated retain and forget classes) (Reviewer kqD5)
- **C4**. Missing comparisons against other methods like influence functions-based approaches or differential privacy (Reviewer kqD5)
- **C5**. Missing discussion of memory and compute complexity, and convergence behaviour, introduced by the meta-learning framework. Especially as the size of the forget set size and diversity increases (Reviewer heUx, Reviewer kNNt)
- **C6**. Missing identification and measurement of unintended consequences of the proposed method, like lack of plasticity (Reviewer heUx)
- **C7**. Missing confidence intervals and statistical significance tests (Reviewer kNNt)
- **C8**. Missing analyses of sensitivity to hyperparameters (Reviewer kNNt)

**Reviewer Concerns:**

The authors have addressed C1 and C7 well, and C5, C6 and C8 partially (see comments below for justification).

The outstanding issues are the lack of convergence analysis (and ideally theoretical foundation), the partially unrealistic experimental setup, missing comparisons with relevant works (see below), and scalability analyses.

This work has various strong points, including the novelty of using meta-learning objectives for proactively addressing unlearning needs, clear writing and well-justified motivation, a model agnostic method, and clear evidence of effectiveness in the settings explored (set of datasets, metrics and baselines compared against).
However, to meet the bar for acceptance, the paper should include missing comparisons, doing so comprehensively across metrics to show relevant trade-offs (between efficiency/amortized cost, unlearning quality, and retention) and across datasets, and address remaining issues brought up by the reviewers about scalability concerns (at least quantifying this) and convergence issues.

**Reviewer Scores:**

**Reviewer kqD5**
The authors partially address C1 by offering a real-world example where organizations possess domain knowledge and operational experience that naturally informs the categorization of data into “stable” vs “revocable”. I find this partially convincing, though it should be noted that this is use-case dependent, as there are other applications for unlearning where we won’t possess that knowledge (e.g. unlearning dangerous knowledge or unwanted behaviours), so it’s important to clarify the scope of unlearning addressed in this work. The authors also offer additional experiments to quantify the degree of misclassification that Ready2Unlearn can tolerate. This helps make the argument that the method can operate even under imperfect knowledge of which data is likely to be requested to be unlearned and is convincing.
The concern about lack of theoretical foundation and convergence analysis has not been addressed well. The authors point to the fact that they borrow the MAML algorithm which they claim has a theoretical foundation. However, they do not perform any convergence analyses in their setting.
The concern about the unrealistic experimental setup was also not addressed very convincingly. The authors state that their intent is essentially a proof of concept that the shift from reactive to proactive methods is worthwhile to further investigate, but their intent is not to “replicate real-world deployment settings”. I don’t find this reasoning convincing. While replicating real-world settings may be beyond the scope of a research paper, there is a plethora of unlearning benchmarks that offer more complex and interesting scenarios than class unlearning, and failing to use any of these is a weakness of the paper.
To address C4, the authors have added comparisons with a differential privacy method, which is a great start, but this might be a relatively weak baseline. Instead of vanilla DP, the authors could compare against DP-like methods specifically designed for unlearning, since they can also be seen as “proactive” rather than “reactive” techniques too. For example, [1], [2] (see references below). Also, it is unclear how the clipping norm and noise values were chosen for DP-SGD. Changing this will give rise to different trade-offs, between the degree of privacy (the larger this is, the faster downstream unlearning will be) and the utility of the model. A systematic study of this trade-off space is needed to make a convincing argument about the superiority of one method over another. If looking just at downstream unlearning efficiency, we can get this to become very high by just doing aggressive DP-SGD (but this will not lead to a useful model).
Notice that other “proactive” techniques also exist, like SISA [3] and newer variants, that modify the architecture itself ahead of time for the purpose of facilitating future unlearning.
(The authors answer other questions that the reviewer brought up in a reasonable way).
Based on the above, I doubt that the reviewer who gave an initial score of 2 would have updated the score substantially.

References
—--------------
[1] Langevin unlearning: A new perspective on noisy gradient descent for machine unlearning. NeurIPS 2024 spotlight.
[2] DP2Unlearning: An Efficient and Guaranteed Unlearning Framework for LLMs (preprint, sharing here just for additional context)
[3] Machine Unlearning. S&P 2021.


**Reviewer heUx**
The authors responded to concern C5 by adding a computational cost analysis section to the appendix where they empirically compare the runtime of standard pretraining to that of the meta-learning training phase introduced by their method. This does partially address the concern. To strengthen the analysis, the authors may consider reporting the amortized overall cost of training and unlearning. Specifically, because the pretraining only needs to happen once, we can amortize the cost of pretraining over the unlearning runs that need to happen later (from the same pretrained model). The authors did not report convergence analysis nor discuss memory complexity in their response.
The authors also add analysis of how forget set diversity (but not size) affects results, again partially addressing the reviewer’s comment.
Finally, the authors conducted new experiments to explore “unintended consequences” in continual learning scenarios, the effect of similarity of the forget and retain set on recovery effects, and other analyses, addressing various questions and suggestions of this reviewer.
This reviewer is very positive about the paper, but it is unclear if a discussion phase between the reviewers would have convinced this reviewer that concerns that others have raised are significant.

**Reviewer kNNt**
The authors address C7 by adding the result of statistical tests in the captions of respective figures.
The concern about scalability to larger models has not been addressed convincingly. The authors acknowledge that this (and other practical concerns) may be an issue, but argue that their work aims for a proof-of-concept and “food for thought” for the community and is not concerned with addressing these practical issues. I don’t find this very convincing as I do feel that it’s the responsibility of the authors to comment on the scalability to larger models and attempt to quantify the cost of their approach.
The authors partially address C1, in the same way as for Reviewer kqD5 (see above).
The authors also conducted other experiments to respond to questions the reviewer raised, including analyses of the overlap between the retain and forget sets and discussed the compatibility of their framework with unlearning algorithms aside from just gradient ascent. They answered concern C8 by reporting how they chose hyperparameters and conducting a sensitivity analysis for one out of the weighing coefficients on MNIST, partially addressing the reviewer’s question.
Overall, due to unresolved concerns, I doubt that this reviewer would have raised their score beyond a weak reject.

---

### Decision · Program_Chairs · 2026-01-26

Reject